# RAD: Retrieval High-quality Demonstrations to Enhance Decision-making

## Abstract

Offline reinforcement learning (RL) learns policies from fixed datasets, thereby avoiding costly or unsafe environment interactions. However, its reliance on finite static datasets inherently restricts the ability to generalize beyond the training distribution. Prior solutions based on synthetic data augmentation often fail to generalize to unseen scenarios in the (augmented) dataset. To address these challenges, we propose Retrieval High-quAlity Demonstrations (RAD) for decision-making, which innovatively introduces a retrieval mechanism into offline RL. Specifically, RAD retrieves high-return and reachable states from the offline dataset as target states, and leverages a generative model to generate sub-trajectories conditioned on these targets for planning. Since the targets are high-return states, once the agent reaches such a target, it can continue to obtain high returns by following the associated high-return actions, thereby improving policy generalization. Extensive experiments confirm that RAD achieves competitive or superior performance compared to baselines across diverse benchmarks, validating its effectiveness. Our code is available at https://anonymous.4open.science/r/RAD_0925_1-690E.

## 1 Introduction

Offline reinforcement learning (RL) aims to learn effective decision policies purely from static datasets, without further interaction with the environment Levine et al. (2020); Prudencio et al. (2023); Park et al. (2024). This setting is essential for domains where active exploration is costly or unsafe, such as robotics Kalashnikov et al. (2021), healthcare Fatemi et al. (2022), and autonomous driving Shi et al. (2021). Despite promising advances, offline RL faces a fundamental limitation: the finite scale of static datasets inherently restricts the learned policy's ability to generalize beyond the training distribution. As illustrated in Figure 1(a), it is challenging to learn a policy that enables the agent reach the target state $G$ from the initial state $S$ using an offline dataset containing only two trajectories. This is because the two trajectories are too far apart, making it difficult for existing offline RL algorithms to generalize to the transition from $S$ to $G$.

To overcome this, recent works typically generate transitions to augment the original dataset, alleviating the negative impact of finite static datasets in offline setting Lu et al. (2023); Li et al. (2024). As shown in Figure 1(b), a new sub-trajectory is generated by the augmentation-based methods, which enable the learning of the policy to support the agent to start from state $S$ and reach state $G$. However, these augmentations are typically generated in a static offline manner, which lack flexibility. Once generated, the augmented dataset remains fixed and cannot adapt to dynamic situations, as shown in Figure 1(c): if the agent later encounters a new state (e.g., a different start state out of the distribution of augmentation and original dataset), there may be no existing demonstration or augmented path that provides meaningful guidance. Consequently, the policy may fail to generalize again, especially under distributional shifts or changing task demands. This highlights the brittleness and limited flexibility of static augmentation methods in offline RL.

A promising approach to achieve effective generalization in offline RL is to adopt an adaptive mechanism: one that adaptively stitches to high-reward trajectories within a certain range to escape out-of-distribution situations or low-reward scenarios. As it is illustrated in Figure 1(d), Starting from a new state S, the agent first plans towards a state located along a high-return trajectory. Once the agent reaches this state, it can then easily navigate to the target state $G$ by leveraging the experience from the high-return trajectory. Inspired by that, we propose Retrieval High-quality Demonstrations

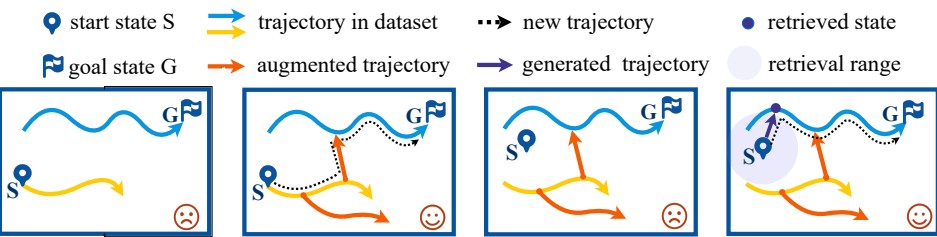

Figure 1: (a) Far-apart trajectories in the offline dataset make it difficult for the policy to learn how to reach $G$ from $S$. (b) By connecting segments, augmentation-based methods expand the dataset and facilitate learning a policy that can reach the $G$ from $S$. (c) When the initial state $S$ falls into an out-of-distribution (OOD) region again, the old augmented data cannot support the agent in learning a policy from $S$ to $G$. (d) RAD dynamically retrieves high-value and reachable states as intermediate targets to guide the agent from $S$ to $G$.

(RAD) in this paper, which is built upon a retrieval mechanism and a generative model. It uses the retrieval mechanism to select states from high-return trajectories in the surrounding region as target states for planning. The agent then leverages the generative model to generate subsequent trajectories toward the target state to achieve higher rewards through interaction with the environment. In such a manner, RAD can efficiently facilitate the transition from OOD states or low-reward regions to potentially high-reward states without relying on complex data augmentation processes. We conduct extensive experiments on D4RL dataset, and the experiment results demonstrate the effectiveness of the RAD.

Our main contributions are: (i) We propose RAD, which retrieves states from high-return trajectories as the target for planning; (ii) RAD can efficiently facilitate the transition from OOD states or low-reward regions to potentially high-reward states without relying on complex data augmentation processes; (iii) The extensive experiments on the widely-used D4RL datasets demonstrate the superiority of RAD.

## 2 RELATED WORK

Offline reinforcement learning (RL) aims to learn decision policies from static datasets without additional environment interaction Levine et al. (2020); Prudencio et al. (2023). The most straightforward solution is behavior cloning (BC), which treats offline RL as a supervised learning problem by directly imitating the behavior policy in the dataset. Another line of work reformulates policy learning as a sequence modeling problem Chen et al. (2021); Janner et al. (2021). For example, Decision Transformer (DT) Chen et al. (2021) conditions on the return-to-go and models entire trajectories with a Transformer, enabling long-horizon planning. More recently, diffusion-based methods Ajay et al. (2022); Janner et al. (2022); Dong et al. (2024) such as Diffuser Janner et al. (2022) apply generative diffusion models to synthesize trajectories, showing strong performance across various offline RL benchmarks. Despite these advances, most of these methods struggle to generalize beyond the distribution of the offline dataset. Conservatism-based approaches Kumar et al. (2020); Yu et al. (2020); Kidambi et al. (2020), such as CQL Kumar et al. (2020) and MOPO Yu et al. (2020), attempt to mitigate extrapolation errors by constraining the learned policy within the support of the dataset, either by penalizing out-of-distribution actions or introducing uncertainty-aware rollouts. However, these methods fundamentally keep the policy restricted to the offline dataset distribution and cannot fully exploit potentially better behaviors outside it. Data augmentation approaches Lu et al. (2023); Li et al. (2024), such as Synthetic Experience Replay (SER) Lu et al. (2023) and DiffStitch Li et al. (2024), enrich the dataset by generating or stitching trajectories, partially alleviating OOD issues. Yet, once trained on the augmented dataset, the policy often fails to adapt to new states beyond the synthesized distribution.

To address the issue, we propose a method called Retrieval High-quAlity Demonstrations (RAD), which retrieves high-return states from the surrounding region and plans toward them to better handle decision-making under OOD conditions, thereby guiding the policy to achieve higher rewards.

## 3 PRELIMINARY

### 3.1 DIFFUSION MODEL

Diffusion Models Sohl-Dickstein et al. (2015); Song et al. (2020); Ho et al. (2020) are the generative models that generate data by progressively removing Gaussian noise. Diffusion models are typically have two processes: forward process that gradually corrupts the data with noise, and a reverse process that learns to reconstruct the original data distribution by removing noise. In the forward process, given a clean sample $\boldsymbol{x} \sim q(\boldsymbol{x})$, diffusion models treat $\boldsymbol{x}$ as the initial sample $\boldsymbol{x}^0$, and inject Gaussian noise step by step with $q(\boldsymbol{x}_t \mid \boldsymbol{x}_{t-1}) = \mathcal{N}(\boldsymbol{x}_t \mid \sqrt{1 - \beta_t}\, \boldsymbol{x}_{t-1}, \beta_t \boldsymbol{I})$, where $\boldsymbol{I}$ is the identity matrix, and $\beta_t$ controls the noise level at step $t$. As the forwarding process progresses, the sample becomes increasingly corrupted by noise. After K steps, sample $\boldsymbol{x}$ is transformed into pure Gaussian noise $\boldsymbol{x}^K$. The reverse process starts from a pure Gaussian noise, it aims to recover $\boldsymbol{x}$ by gradually removing the noise step by step with $p_\theta(\boldsymbol{x}_{t-1} \mid \boldsymbol{x}_t) = \mathcal{N}(\boldsymbol{x}_{t-1} \mid \mu_\theta(\boldsymbol{x}_t, t), \Sigma_\theta(\boldsymbol{x}_t, t))$, where the mean can be re-expressed with $\mu_\theta(\boldsymbol{x}_t, t) = \frac{\sqrt{\alpha_t}(1 - \bar{\alpha}_t)}{1 - \bar{\alpha}_{t-1}} \boldsymbol{x}_t + \frac{\sqrt{\bar{\alpha}_{t-1}}\beta_t}{1 - \bar{\alpha}_t} \phi_\theta(\boldsymbol{x}_t, t)$, with $\alpha_t = 1 - \beta_t$ and $\bar{\alpha}_t = \prod_{s=1}^{t} \alpha_s$, $\phi_\theta$ is model to reconstruct $\boldsymbol{x}$. Fixing $\Sigma_\theta(\boldsymbol{x}_t, t) = \beta_t I$ Ho et al. (2020), the learning objective is formulated by minimizing the mean squared error between the true signal and the model prediction:

$$\mathcal{L} = \mathbb{E}_{\boldsymbol{x},\, t \sim [1,T]} \left[ \| \boldsymbol{x}^0 - \psi_\theta(\boldsymbol{x}_t, t) \|^2 \right]. \tag{1}$$

### 3.2 PROBLEM DEFINITION

RL is typically formulated as a Markov Decision Process (MDP). Formally, a MDP is given by $\mathcal{M} = \{\boldsymbol{S}, \boldsymbol{A}, P, r, \gamma\}$, where $\boldsymbol{S}$ is the state space, $\boldsymbol{A}$ is the action space, $P$ is the transition function, $r$ is the reward function, and $\gamma \in (0, 1)$ is the discount factor. At each timestep $t$, the agent observes the environment state $\boldsymbol{s}_t$, takes an action $\boldsymbol{a}_t$ according to a policy $\pi_\theta$ parameterized by $\theta$, then receives an instantaneous reward $r_t$ from environment, and transits to state $\boldsymbol{s}_{t+1}$ via $P(\boldsymbol{s_{t+1}} \mid \boldsymbol{s_t}, \boldsymbol{a_t})$. The interaction history is represented as a trajectory $\tau = \{(\boldsymbol{s_t}, \boldsymbol{a_t}, r_t) \mid t \geq 0\}$. We define the cumulative discounted reward from step $t$ as $v_t = \sum_{i \geq t} \gamma^{i-t} r_i$, and refer to it as the return of state $\boldsymbol{s_t}$. Additionally, the return of a complete trajectory $\tau$ is defined as $R(\tau) = \sum_{t \geq 0} \gamma^t r_t$.

We focus on the offline RL setting, where the agent cannot interact with the environment and must learn from a fixed dataset $\mathcal{D} = \{\tau_i\}_{i=1}^{N}$ consisting of $N$ trajectories collected by some unknown behavior policy. Each trajectory $\tau_i$ is a sequence of state-action-reward tuples: $\tau_i = \{(\boldsymbol{s_0}, \boldsymbol{a_0}, r_0), (\boldsymbol{s_1}, \boldsymbol{a_1}, r_1), \dots, (\boldsymbol{s_{T-1}}, \boldsymbol{a_{T-1}}, r_{T-1})\}$, where $T$ denotes the length of each trajectory. Our goal is to learn a policy $\pi_\theta$ that maximizes the expected return without interacting with the environment:

$$\pi_\theta = \arg \max_\theta \mathbb{E}_{\tau \sim \pi_\theta}[R(\tau)]. \tag{2}$$

## 4 METHOD

We propose a method called Retrieval High-quAlity Demonstrations (RAD) for offline RL, which integrates a retrieval augmented mechanism with sub-trajectory generation to improve policy generalization under the scenarios beyond the dataset coverage. As it is illustrated in Figure 2, RAD is composed of a target selection (TS) module, a step estimation (SE) module, and a planning (PL) module. Given the current state $\boldsymbol{s}_t$, TS first retrieves reachable and high-return states as targets as $\boldsymbol{s}_t^g$. SE then estimates the step $\hat{i}_t$ transit from the current state $\boldsymbol{s}_t$ to the target state $\boldsymbol{s}_t^g$. PL finally randomly initializes a noisy sub-trajectory, and offsets $\boldsymbol{s}_t$ and $\boldsymbol{s}_t^g$ to the first position and $\hat{i}_t$ position, generating the subsequent trajectory and making a decision. Since the targets are high-return states, once the agent reaches such a target state, it can obtain a high return by following the high-return action associated with the target state, thereby addressing the generalization of policy in low-return or OOD regions. In the following, we will discuss the TS and SE first, and subsequently the PL.

### 4.1 TARGET SELECTION(TS) MODULE

The target selection (TS) module aims to dynamically retrieve and select a high-return and reachable target state $\boldsymbol{s}_t^g$ from the offline dataset for the subsequent planning. To conduct that, we first construct

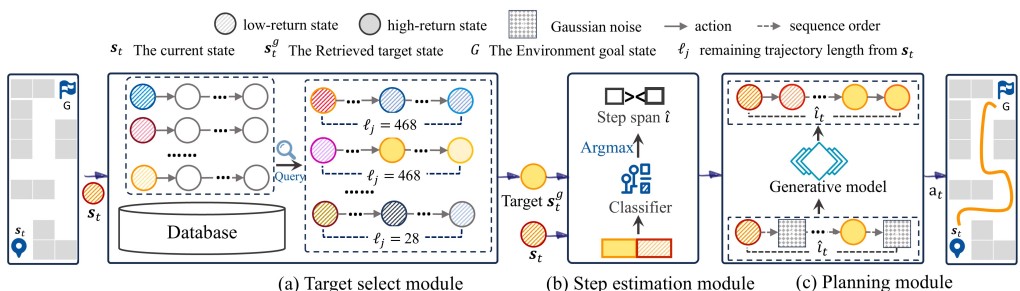

Figure 2: Overall framework of RAD.

a database that contains all states from expert trajectories (please refer to the Appendix D for more details). Each entry in the database is composed of: (1) state $s_i$: the feature vector representing the environment state at timestep $i$; (2) trajectory ID: the identifier that indicates which trajectory the state $s_i$ belongs; (3) Timestep $i$: the index of state $s_i$ within its trajectory; (4) Discounted return $v_i$ : the cumulative discounted return starting from state $s_i$.

Based on the current state $s_t$ and the database, we take the following steps to obtain the target state:

**Selecting the similar states:** Given the current state $s_t$, we first use it as a query vector to retrieve states from the database based on their similarity to $s_t$. We employ two metrics to measure state similarity. For locomotion and manipulation tasks, the similarity between the current state $s_t$ and database states is computed using the cosine similarity. For navigation tasks, the similarity is measured by the Euclidean distance in the two-dimensional spatial plane. We then select the top-$k$ states with similarity greater than $\delta$ and include them in the set $\mathcal{C}_s$.

**Extracting the high-return states:** For each state $s_i \in \mathcal{C}_s$, we consider the subsequent trajectory within the next $H - 1$ steps to ensure a sufficient number of candidate states, and compute the cumulative discounted return starting from each state. Specifically, for any state $s_j$ with $i \leq j \leq H-1$ in the subsequent trajectory of $s_i$, its cumulative discounted return is denoted as $v_j$. We then extract those states whose return lies within a tolerance threshold $\eta$ of the highest return $v^*$ :

$$|v_j - v^*| \leq \eta, \tag{3}$$

where $v^*$ denotes the maximum return among the candidate states in the batch. The states satisfying this condition are collected into the set $\mathcal{C}_g$.

**Drawing out the trajectory-continuable state:** To improve planning robustness and provide a richer context for the trajectory generation module, we prioritize candidates associated with longer remaining trajectories in the original demonstrations. Concretely, let $\ell_j$ denote the length of the remaining trajectory starting from $s_j$, $s_j \in \mathcal{C}_g$, we select the target state with:

$$s_t^g = \arg \max_{s_j \in \mathcal{C}_g} \ell_j. \tag{4}$$

The selected target state $s_t^g$ is then used to guide downstream planning.

## 4.2 STEP ESTIMATION(SE) MODEL

While the retrieved target state $s_t^g$ specifies the direction the agent should move toward for high return, but does not indicate the temporal distance, *i.e.*, the number of steps needed to reach the target state from the current state, making it difficult for the agent to plan effectively. From the perspective of Markov Decision Processes and conditional generative modeling, capturing temporal alignment is crucial Ajay et al. (2022): without modeling the expected arrival step, the generated sub-trajectories are prone to overshooting, stalling, or degenerating, leading to incoherent or infeasible behaviors. Therefore, we design the SE module to predict the step span between the current state $s_t$ and the target state $s_t^g$. However, directly predicting the step span is a discrete regression problem, which is difficult than classification Xiong & Yao (2022). To alleviate this difficulty, we reformulate the

regression problem as a classification task. Specifically, we concatenate the current state $s_t$ with the target state $s_t^g$, and feed the concatenation to a multilayer perceptron $f_e$:

$$e_t = \text{Sigmoid}(f_e([s_t, s_t^g])), \tag{5}$$

where $[.,.]$ denotes the concatenation operation, and $e_t$ is a $H - 1$-dimensional vector, the i-th dimension represents the probability of $s_t$ requires $i$ steps to reach $s_t^g$. Then, we obtain the estimated step span with :

$$\hat{i} = \arg\max e_t. \tag{6}$$

## 4.3 PLANNING(PL) MODULE

With the current state $s_t$, target state $s_t^g$ and the estimated step count $\hat{i}$, the PL module aims to apply a diffusion model to generate the subsequent trajectory for planning. Specifically, we randomly initialized a noisy sub-trajectory $\tau_{temp}$ with length of $H(H \geq \hat{i})$:

$$\tau_{temp} = \{\hat{\psi}_t^K, \hat{\psi}_{t+1}^K, \ldots, \hat{\psi}_{t+H}^K\}, \tag{7}$$

where each element $\hat{\psi}_t^K$ in $\tau_{temp}$ represents either a noisy state-action pair ($\hat{\psi}_t^K = \{\hat{s}_t^K, \hat{a}_t^K\}$)) or a noisy state ($\hat{\psi}_t^K = \hat{s}_t^K$) only, and $K$ denotes the diffusion steps. Then, we obtain $\hat{\tau}_t^K$ from $\tau_{temp}$ by substituting the state in $\hat{\psi}_t^K$ with the current state $s_t$, and the state in $\hat{\psi}_{\hat{i}}^K$ with the target state $s_t^g$.

Starting from $\hat{\tau}_t^K$, we conduct the reverse denoising process of the diffusion model Janner et al. (2022) to obtain a clean sub-trajectory. Each denoising step is parameterized as:

$$p_\theta(\hat{\tau}_t^{k-1} | \hat{\tau}_t^k) = \mathcal{N}(\mu_\theta(\hat{\tau}_t^k, k) + \rho\nabla\mathcal{J}_\phi(\hat{\tau}_t^k), \beta_k I) , \tag{8}$$

$$\mu_\theta(\hat{\tau}_t^k, k) = \frac{\sqrt{\alpha^k}(1 - \bar{\alpha}^{k-1})}{1 - \bar{\alpha}^{k-1}}\hat{\tau}_t^i + \frac{\sqrt{\bar{\alpha}^{k-1}}\beta^k}{1 - \bar{\alpha}^k}\hat{\tau}_t^{k,0} . \tag{9}$$

Here $\hat{\tau}_t^{k,0} = \phi_\theta(\hat{\tau}_t^k, k)$ represents the $\tau_t^0$ constructed from $\hat{\tau}_t^k$ at diffusion step $k$, $\phi_\theta(\cdot, \cdot)$ is a network for trajectory generation, $k \sim [1, K]$ is the diffusion step, $\rho$ is a scaling factor controlling the guidance strength, $\mathcal{J}_\phi(\cdot)$ predicts the trajectory return to provide guidance for generation. $I$ denotes the identity matrix, and $\beta_i$ is the noise schedule coefficient that determines the proportion of noise injected at denoising step $i$. After $K$ denoising steps, we obtain the generated sub-trajectory $\hat{\tau}_t^0 = \{\hat{\psi}_t^0, \hat{\psi}_{t+1}^0, \ldots, \hat{\psi}_{t+H}^0\}$.

If the element in $\tau_{temp}$ is composed of a noisy state-action pair, $\hat{\tau}_t^0$ is the sequence of clean state-action pairs, we directly take the action in $\hat{\psi}_{t+1}^0$ to interact with the environment. If the element in $\tau_{temp}$ is composed of noisy states only, $\hat{\tau}_t^0$ is the sequence of clean states, and we then take out the state in $\hat{\psi}_{t+1}^0$, and feed it with the current state $s_t$ to a inverse dynamic model to obtain the action to interact with the environment:

$$a_t = f_a(s_t, \hat{s}_{t+1}), \tag{10}$$

where $\hat{s}_{t+1}$ denotes the generated states for step $t + 1$, $f_a$ is the inverse dynamic model.

Considering existing diffusion-based offline RL methods have already demonstrated strong quality in generating subsequent trajectories, and we focus on improving offline RL through a retrieval-augmented mechanism rather than trajectory generation itself, we select Diffuser Janner et al. (2022) and DiffuserLite Dong et al. (2024), two diffusion-based but totally different methods[1], to conduct the generation in implementation Eq. (8). Correspondingly, we have two variants: (1) **D-RAD**, which integrates our retrieval-augmented mechanism with the trajectory generation of Diffuser, producing trajectories of states and actions for decision making; (2) **DL-RAD**, which integrates our retrieval-augmented mechanism with the trajectory generation of DiffuserLite, producing trajectories of states only, after which actions are predicted using an inverse dynamics model for decision making.

---

[1]Our retrieval-augmented mechanism is, in theory, compatible with any trajectory–generation–based offline RL algorithm

### 4.4 MODEL LEARNING

Our method is trained with three losses: (1) the generation loss, which constrains the generation of planning toward the high-return states; (2) the generation guidance loss, which constrains the guidance function; (3) the step estimation loss, which guarantees the accuracy of step estimation.

**Generation loss.** To train the generation of planning toward high-return states, we employ a pseudo target strategy. Concretely, we first sample a demonstration trajectory from the offline dataset:

$$\tau_t^0 = \{\psi_t^K, \psi_{t+1}^K, \ldots, \psi_{t+H-1}^K\}, \tag{11}$$

where $\psi_t^K$ denotes the vector representation of the state (the state-action pair if the planning module is used to generate the state-action pair) of step $t$. Then, we randomly select an offset $i \sim \mathcal{U}(1, H-1)$ and set the vectors in $\psi_{t+i}^K$ as the pseudo target and applying forward diffusion to the sub-trajectory, the denoising network $\phi_\theta$ is trained by minimizing the noise prediction error:

$$\mathcal{L}_d = \mathbb{E}_{\boldsymbol{\tau}_t \in \mathcal{D}, t>0, k \sim [1,K]} \left[ \|\boldsymbol{\tau}_t - \phi_\theta(\hat{\boldsymbol{\tau}}_t^k, k)\|^2 \right], \tag{12}$$

**Generation guidance loss.** The generation guidance $\mathcal{J}_\phi(\cdot)$ is optimized by minimizing the mean squared error between the predicted trajectory return signal and the ground-truth return signal over the offline dataset $\mathcal{D}$:

$$\mathcal{L}_g = \mathbb{E}_{\tau \sim \mathcal{D}} \left[ (\mathcal{J}_\phi(\tau) - C(\tau))^2 \right]. \tag{13}$$

For D-RAD, $C(\tau)$ corresponds to the cumulative discounted return of the trajectory $R(\tau)$. For DL-RAD, $C(\tau) = \sum_{t=0}^{H-2} \gamma^t r_t + \gamma^{H-1} V(\boldsymbol{s}_{H-1})$, where $V(\boldsymbol{s}_t) = \max \mathbb{E}_\pi \left[ \sum_{\tau=t}^{\infty} \gamma^{\tau-t} r_\tau \right]$ denotes the optimal value function and can be estimated by a neural network through various offline RL methods. Here, $H$ is the temporal horizon.

Step estimation loss. The step estimation loss is formulated as the cross-entropy loss between the predicted step distribution $\boldsymbol{e}_t$ and the ground-truth step count $i$:

$$\mathcal{L}_e = -\mathbb{E}_{(\boldsymbol{s}_t, \boldsymbol{s}_t^g)} \left[ \log \boldsymbol{e}_t[i] \right], \tag{14}$$

where $i$ is the ground truth offset of steps from $\boldsymbol{s}_t$ to $\boldsymbol{s}_t^g$, and $\boldsymbol{e}_t[i]$ denotes the predicted probability for class $i$.

$\mathcal{L}_d$, $\mathcal{L}_g$ and $\mathcal{L}_e$ are optimized independently. The details of the training and testing process are presented in Appendix C.

## 5 EXPERIMENT DESIGN AND RESULTS ANALYSIS

We explored the performance of RAD on a variety of offline RL tasks, including locomotion tasks (HalfCheetah, Hopper, Walker2d), navigation tasks (AntMaze, Maze2d), and manipulation tasks (Kitchen), and aimed to answer the following research questions (RQs): (1) How does RAD perform compared with baseline methods across different environments? (2) Can RAD generalize to new states not covered in the training dataset? (3) How does the key component contribute to the performance of RAD? (4) Are the target states generated by RAD feasible and achievable in practice, and do they provide effective guidance for reaching the final goal?

### 5.1 EXPERIMENT SETTINGS

**Environments and Datasets.** We evaluate the algorithm on various offline RL environments, including locomotion in Gym-MuJoCo Brockman et al. (2016), long-horizon navigation in Antmaze Fu et al. (2020), real-world manipulation in FrankaKitchen Gupta et al. (2019), and 2D navigation tasks in Maze2D Fu et al. (2020). We train models using publicly available datasets (see appendix B for further details).

**Baselines.** To evaluate our RAD, we compare it against a diverse representative offline RL algorithms. These include imitation learning methods such as Behavior Cloning (BC); model-free offline reinforcement learning approaches, including Conservative Q-Learning (CQL) Kumar et al. (2020) and

Table 1: The Performance across benchmark environments[2]. The results correspond to the mean over 3 random seeds with standard errors. Scores within 5% of the maximum per task ($\geq 0.95 \times$ MAX) are highlighted in **bold**. We abbreviate Diffuser as Diff and DiffuserLite as Lite for brevity.

| Dataset | Env | BC | CQL | IQL | MOPO | MOReL | DT | SER | DStitch | DS | DD | Diff | Lite | D-RAD | DL-RAD |
|---|---|---|---|---|---|---|---|---|---|---|---|---|---|---|---|
| Medium-Expert | HalfCheetah | 55.2 | **91.6** | 86.7 | 63.3 | 53.3 | 86.8 | 88.9 | **94.4** | **95.7** | 90.6 | 88.9 | 88.5 | 84.9 ± 0.5 | 90.1 ±0.1 |
| | Hopper | 52.5 | 105.4 | 91.5 | 23.7 | **108.7** | **107.6** | **110.4** | **110.9** | **107.0** | **111.8** | 103.3 | **111.6** | **112.3 ± 0.3** | **110.0 ± 0.3** |
| | Walker2d | **107.5** | **108.8** | **109.6** | 44.6 | 95.6 | **108.1** | **111.7** | **111.6** | **108.0** | **108.8** | 106.9 | **107.1** | **108.1 ±0.1** | **110.2 ±0.2** |
| Medium | HalfCheetah | 42.6 | 44.0 | **47.4** | 42.3 | 42.1 | 42.6 | **49.3** | **49.4** | 47.8 | **49.1** | 42.8 | **48.9** | 44.2 ±0.2 | **48.8 ±0.6** |
| | Hopper | 52.9 | 58.5 | 66.3 | 28.0 | 95.4 | 67.6 | 66.6 | 71.0 | 76.6 | 79.3 | 74.3 | **100.9** | 82.5 ±2.3 | **101.0 ±1.1** |
| | Walker2d | 75.3 | 72.5 | 78.3 | 17.8 | 77.8 | 74.0 | **85.9** | 83.2 | 83.6 | 82.5 | 79.6 | **88.8** | 82.8 ±0.7 | **89.4 ±0.2** |
| Medium-Replay | HalfCheetah | 36.6 | 45.5 | 44.2 | **53.1** | 40.2 | 36.6 | 46.6 | 44.7 | 41.0 | 39.3 | 37.7 | 41.6 | 41.2 ±0.1 | 44.4 ±0.1 |
| | Hopper | 18.1 | 95.0 | 94.7 | 67.5 | 93.6 | 82.7 | **102.4** | **102.1** | 89.5 | **100.0** | 93.6 | 96.6 | **98.0 ±0.6** | **100.4 ±0.4** |
| | Walker2d | 26.0 | 77.2 | 73.9 | 39.0 | 49.8 | 66.6 | 85.7 | 86.6 | 80.7 | 75.0 | 70.6 | **90.2** | 77.6 ±1.2 | **93.5 ±1.2** |
| | Average | 51.9 | 77.6 | 77.0 | 42.1 | 72.9 | 74.7 | 83.1 | **83.8** | 81.1 | 81.8 | 77.5 | **86.0** | 81.3 | **87.5** |
| Play | Antmaze-Medium | 0.0 | 65.8 | 65.8 | 0.0 | 0.0 | 0.0 | 41.0 | 65.8 | 0.0 | 8.0 | 6.7 | 78.0 | 40.0 ± 5.2 | **86.7± 3.6** |
| | Antmaze-Large | 0.0 | 20.8 | 42.0 | 0.0 | 0.0 | 0.0 | 72.9 | 42.0 | 0.0 | 0.0 | 17.3 | 72.0 | 13.3 ± 3.6 | **80.0± 4.2** |
| Diverse | Antmaze-Medium | 0.0 | 67.3 | 73.8 | 0.0 | 0.0 | 0.0 | 40.9 | 73.8 | 0.0 | 4.0 | 2.0 | **92.4** | 6.7 ± 2.6 | **93.3± 2.6** |
| | Antmaze-Large | 0.0 | 20.5 | 30.3 | 0.0 | 0.0 | 0.0 | 37.5 | 30.3 | 0.0 | 0.0 | 27.3 | 68.0 | 26.7 ± 4.7 | **73.3± 4.7** |
| | Average | 0.0 | 43.6 | 53.0 | 0.0 | 0.0 | 0.0 | 48.1 | 53.0 | 0.0 | 3.0 | 13.3 | 77.6 | 21.7 | **83.3** |
| Kitchen | Mixed | 51.5 | 52.4 | 51.0 | 17.3 | 0.0 | 25.8 | 56.1 | 51.0 | 1.6 | 65.0 | 52.5 | **73.6** | 63.3 ± 1.1 | **72.7± 1.4** |
| | Partial | 38.0 | 50.0 | 46.3 | 6.7 | 35.5 | 31.4 | 37.4 | 63.3 | 1.6 | 57.0 | 55.7 | **74.4** | 65.0 ± 1.3 | **71.5± 1.7** |
| | Average | 44.8 | 51.2 | 48.7 | 12.0 | 17.8 | 28.6 | 46.8 | 57.2 | 1.6 | 61.0 | 54.1 | **74.0** | 64.2 | **72.1** |
| Maze2d | Large | 5.0 | 12.5 | 59.0 | -0.5 | 14.1 | 35.7 | 61.7 | 59.0 | **171.6** | 111.8 | 123.0 | 39.1 | 149.2 ± 7.5 | 44.3 ± 9.2 |
| | Medium | 30.3 | 5.0 | 32.8 | 19.1 | 68.5 | 31.7 | 34.1 | 50.2 | 111.7 | 103.7 | 121.5 | 32.2 | **128.2± 6.6** | 78.3 ± 10.4 |
| | U-Maze | 3.8 | 5.7 | 37.4 | -15.4 | 76.4 | 18.1 | 40.5 | 77.0 | 111.3 | 113.8 | 113.9 | 31.2 | **127.4± 1.2** | 78.2± 14.8 |
| | Average | 13.0 | 7.7 | 43.1 | 1.1 | 53.0 | 28.5 | 45.4 | 62.1 | **131.5** | 109.8 | 119.5 | 34.2 | **134.9** | 66.9 |

Implicit Q-Learning (IQL) Kostrikov et al. (2021); model-based methods such as Model-based Offline Policy Optimization (MOPO) Yu et al. (2020) and Model-based Offline Reinforcement Learning (MOReL) Kidambi et al. (2020); return-conditioned methods such as Decision Transformer (DT) Chen et al. (2021); data-augmented methods Synthetic experience replay (SER) Lu et al. (2023) and DiffserStitch Li et al. (2024); and diffusion-based planning methods including Diffuser Janner et al. (2022), Decision stacks Zhao & Grover (2023), Decision Diffuser (DD) Ajay et al. (2022), and the recently proposed DiffuserLite Dong et al. (2024).

**Implementation Details.** For D-RAD, we follow the same planning horizon as Diffuser, while DL-RAD uses the horizon defined in DiffuserLite. The step estimation model $f_e$ is implemented as a 4-layer MLP. More details about hyperparameter please refer Appendix D. Training was conducted on 4 NVIDIA A40 GPUs, an Intel Gold 5220 CPU, and 504GB memory, optimized with the Adam optimizer Kingma & Ba (2014). The baselines are implemented following their official implementations for a fair comparison.

## 5.2 PERFORMANCE

To evaluate the effectiveness of the proposed RAD framework, we compare D-RAD and DL-RAD with representative baselines across different categories on D4RL. Results in Table 1 show that RAD achieves the best or near-best performance on 16 out of 18 datasets (RQ1). More specifically, (1) in the MuJoCo environments, on sub-optimal datasets (Medium and Medium-Replay), DL-RAD exhibits more pronounced improvements compared to existing methods. Specifically, on the Walker-Medium-Replay dataset, RAD outperforms the highest-scoring baseline, DiffuserLite, by approximately 3. This improvement can be attributed to RAD's retrieval of *high-return and reachable* states from the offline dataset as target states. In sub-optimal datasets with heterogeneous data quality, many low-return or sub-optimal trajectories exist, which may mislead conventional methods. By retrieving high-quality state segments, RAD effectively skips low-value regions, allowing the policy to learn more high-return behaviors during training, thereby improving performance. In contrast, on the Medium-Expert datasets, most trajectories are already near-optimal, and even without the retrieval mechanism, policies can learn high-return behaviors, resulting in limited marginal gains from retrieval augmentation; (2) in the AntMaze environments, RAD consistently outperforms all baseline methods across different datasets. For example, on Antmaze-Medium-Play, DL-RAD achieves a score of 86.7, surpassing the best-performing baseline, DiffuserLite (78.0), by approximately 8.7. On

---

[2]Results for SER and DStitch are obtained by applying the methods with IQL as the offline RL algorithm.

Table 2: Performance under distribution shifts. Models are trained on Medium-Replay datasets and evaluated with initial states replaced by states from the corresponding Random datasets. The best results are in bold.

| Dataset | CQL | DT | MOPO | DiffStitch | DiffuserLite | DL-RAD |
|---|---|---|---|---|---|---|
| HalfCheetah | 37.9 | 26.3 | **62.3** | 26.4 | 37.6 | 39.2 |
| Hopper | 63.6 | 35.7 | 39.5 | 37.6 | 60.4 | **90.2** |
| Walker2d | 50.6 | 46.0 | 78.6 | 13.7 | 73.9 | **85.8** |

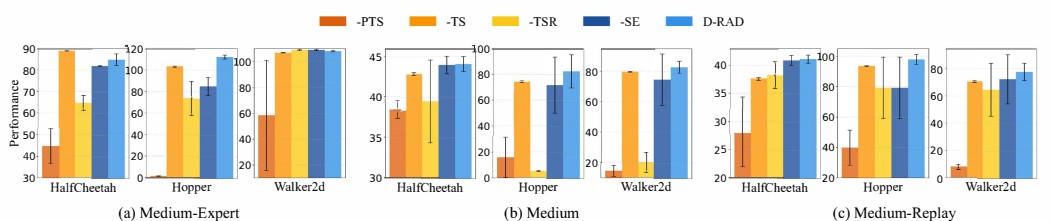

Figure 3: Results of ablation experiments on different variants.

Antmaze-Large-Play, DL-RAD reaches 80.0, which is more than 7 higher than other methods. This indicates that RAD can effectively perform long-horizon planning under sparse reward conditions. By selecting high-return target states from expert trajectories and generating feasible action sequences, RAD guides the agent along a reasonable path toward the final goal; (3) in Maze2d, both D-RAD and DL-RAD surpass Diffuser and DiffuserLite, demonstrating that the retrieval-guided mechanism helps generate higher-quality long-horizon action sequences.

## 5.3 GENERALIZATION

Offline reinforcement learning faces the critical challenge of whether the learned policy can generalize to situations not present in the training dataset. To evaluate this, we conduct experiments in the Medium-Replay. Specifically, we first initialize the states by randomly sampling from the corresponding Random dataset. Subsequently, we leverage policies pre-trained on the Medium-Replay dataset to interact with the environment. As shown in Table 2, DL-RAD demonstrates clear improvements over DiffuserLite and other baselines in Hopper and Walker2d, while underperforming MOPO in HalfCheetah. We hypothesize that this performance gap arises because the HalfCheetah Medium-Replay dataset exhibits both higher average cumulative returns and richer trajectory diversity compared to Hopper and Walker2d Shan et al., allowing MOPO to fully exploit them through dynamics modeling and thereby achieve superior performance. In this case, DL-RAD's retrieval-augmented mechanism provides limited additional benefits compared with dynamics modeling of MOPO. However, in Hopper and Walker2d, DL-RAD achieves substantially higher returns than all other baselines. This suggests that the retrieval-augmented target states enable the agent to better exploit trajectories in the offline dataset for decision making, thereby allowing the learned policy to generalize to new states not covered in the training dataset (RQ2).

## 5.4 ABLATION STUDY

To evaluate the contribution of each component in the RAD, we conduct ablation studies. Specifically, we have three variants:

- **-TS** removes the TS module.

- **-TSR** removes the TS module and randomly samples targe states from the dataset.

- **-SE** removes the SE module and replaces the predicted transition horizon with a randomly selected number of steps.

- **-PTS** replaces the random offset $i \sim U(1, H-1)$ in the pseudo target strategy (4.4) with a fixed step $i$, so that the pseudo target is always selected at the same horizon within the trajectory.

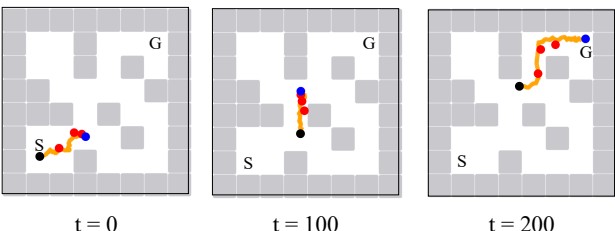

Figure 4: Visualization of policy predictions and real environment roll-outs in the AntMaze environment. Black dots denote the start states of the corresponding steps, blue dots indicate the target states ($S$ for the map's starting position and $G$ for the final goal), red dots represent the intermediate states, and the orange line represents the actual trajectory of agent-environment interaction over time, starting from the current moment.

Specifically, we observe the following : (1) Compared **-TSR**, **-TS** and **D-RAD**, **-TSR** performs worst, **-TS** is better, and **D-RAD** achieves the best performance in most cases. This trend confirms that both the TS and SE modules contribute positively (RQ3), as removing them (**-TS**) reduces performance, and further replacing TS with random targets (**-TSR**) degrades it even more. (2) The contrast between **-TSR** and **D-RAD** highlights that retrieving appropriate target states is crucial. While randomly injected targets introduce noise and mislead the policy, retrieved targets provide informative guidance that effectively directs the agent toward high-return regions. (3) **-PTS** performs worse than **D-RAD**. This validates the pseudo target strategy: a fixed horizon limits adaptability, whereas sampling random offsets across horizons enriches training and improves robustness.

The additional ablation study results for **DL-RAD** are provided in Appendix E.2.

### 5.5 Visulization

To further investigate the target states generated by RAD, we conducted a visualization experiment. Specifically, we selected the AntMaze-medium-replay environment and visualized part of the target states generated by the DL-RAD agent, along with the actual trajectories obtained through environment interactions.

The results are shown in Fig. 4. From the figure, we can observe the following (RQ4): (1) The target states generated by the policy are located at reasonable positions and do not lead the agent to collide with or pass through walls, indicating that the target states are reasonable. (2) Guided by these target states, the agent can successfully reach the final goal $G$, demonstrating that enhancing the decision-making with the guidance of target states is effective. (3) The actual trajectories obtained from environment interactions align well with the generated target states, suggesting that the targets are not only theoretically reasonable but also practically achievable, thereby validating the reachability of our method.

## 6 Conclusion and Discussion

We presented RAD, a retrieval-augmented method for offline RL. RAD improves generalization by dynamically retrieving high-return states as target states and leveraging diffusion-based trajectory generation for planning. By conditioning on these target states, the agent is guided toward reachable high-return regions, gradually escaping low-return and poorly covered areas, and thereby generalizing to previously unseen states. Experiments on D4RL tasks show that RAD matches or outperforms prior methods across diverse settings. This demonstrates the potential of our retrieval-augmented mechanism in overcoming data coverage limitations in offline RL. However, RAD still relies on the coverage of the offline dataset. When long-horizon target states are absent, the generated trajectories may become suboptimal, thereby affecting planning performance.

Our current experiments are primarily conducted on standard D4RL tasks, with visualizations and analyses limited to a few environments. Future work could extend RAD to more complex and diverse scenarios. In addition, we plan to investigate more efficient retrieval strategies to further improve the applicability and effectiveness of RAD.

## 7 ETHICS STATEMENT

This research was conducted in accordance with established ethical standards for scientific work. Topics considered include, but are not limited to, the involvement of human subjects, dataset usage and release practices, potentially harmful insights, research methodologies and applications, conflicts of interest and sponsorship, discrimination/bias/fairness concerns, privacy and security issues, legal compliance, and research integrity (e.g., IRB approvals, documentation, and research ethics). Specifically, our study does not involve human subjects or personally identifiable information, and therefore no Institutional Review Board (IRB) approval was required. All datasets used are publicly available and released under appropriate licenses. Potential risks, including fairness, bias, privacy, and unintended harmful use of the findings, were carefully assessed, and steps were taken to minimize such risks. We affirm that our work complies with research integrity guidelines, including accurate reporting, transparency, and reproducibility.

## 8 REPRODUCIBILITY STATEMENT

We have taken multiple steps to ensure the reproducibility of our results. The main text provides detailed descriptions of the model architecture and training procedure, while the appendix includes additional explanations of implementation details and hyper-parameters. All datasets used in our experiments are publicly available. Furthermore, we release the source code in an anonymous repository, enabling researchers to faithfully reproduce our experiments.

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

## A    LARGE LANGUAGE MODELS USAGE

Large Language Models (LLMs) were used solely for language polishing and minor stylistic improvements. They did not contribute to the conceptual development, methodology, experiments, or analysis of this work. The authors take full responsibility for all content.

## B    DETAILS OF EXPERIMENTAL SETUP

Gym-MuJoCo Brockman et al. (2016) on D4RL consists of three popular offline RL locomotion tasks (Hopper, HalfCheetah, Walker2d). These tasks require controlling three MuJoCo robots to achieve maximum movement speed while minimizing energy consumption under stable conditions. D4RL provides three different quality levels of offline datasets: "medium" containing demonstrations of medium-level performance, "medium-replay" containing all recordings in the replay buffer observed during training until the policy reaches medium performance, and "medium-expert" which combines medium and expert level performance equally. We further analyze the returns distribution of these datasets, showing the differences in trajectory quality among the Medium, Med-Replay, and Med-Expert datasets for HalfCheetah, Hopper, and Walker2d (Figure 5).

FrankaKitchen Gupta et al. (2019) requires controlling a realistic 9-DoF Franka robot in a kitchen environment to complete several common household tasks. In offline RL testing, algorithms are often evaluated on "partial" and "mixed" datasets. The former contains demonstrations that partially solve all tasks and some that do not, while the latter contains no trajectories that completely solve the tasks. Therefore, these datasets place higher demands on the policy's "stitching" ability. During testing, the robot's task pool includes four sub-tasks, and the evaluation score is based on the percentage of tasks completed.

AntMaze Fu et al. (2020) requires controlling the 8-DoF "Ant" quadruped robot in MuJoCo to complete maze navigation tasks. In the offline dataset, the robot only receives a reward upon reaching the endpoint, and the dataset contains many trajectory segments that do not lead to the endpoint, making it a difficult decision task with sparse rewards and a long horizon. The success rate of reaching the endpoint is used as the evaluation score, and common model-free offline RL algorithms often struggle to achieve good performance.

Maze2D Fu et al. (2020) is a navigation task in which a 2D agent needs to traverse from a randomly designated start location to a fixed goal location where a reward of 1 is given. No reward shaping is provided at any other location. The objective of this task is to evaluate the ability of offline RL algorithms to combine previously collected sub-trajectories in order to find the shortest path to the evaluation goal. Three maze layouts are available: "umaze", "medium", and "large". The expert data for this task is generated by selecting random goal locations and using a planner to generate sequences of waypoints that are followed by using a PD controller to perform dynamic tracking.

## C    ALGORITHMS.

---

**Algorithm 1** Training

---

**Require:** Offline dataset $\mathcal{D}$, batch size $B$, diffusion model $\phi_\theta$, step estimation model $f_e$, batch size $B$, diffusion steps $K$
  1: **for** each training iteration **do**
  2:      Sample a batch of trajectories $\{\tau_t\}_{t=1}^B$ from $\mathcal{D}$
  3:      **for** each trajectory $\tau_t$ in batch **do**
  4:          Randomly select an offset $i \sim \mathcal{U}(1, H-1)$
  5:          Set $\boldsymbol{s}_{t+i}$ as pseudo target
  6:          Apply forward diffusion on sub-trajectory $\tau_t$ to obtain noisy $\hat{\tau}_t^k$
  7:          Compute generation loss: $\mathcal{L}_d$ by Eq. 12.
  8:          Compute guidance loss: $\mathcal{L}_g$ by Eq. 13.
  9:          Compute step estimation loss: $\mathcal{L}_e$ by Eq. 14.
 10:      **end for**
 11:      Update $\phi_\theta$ and $f_e$ using gradients from $\mathcal{L}_d, \mathcal{L}_g, \mathcal{L}_e$
 12: **end for**

---

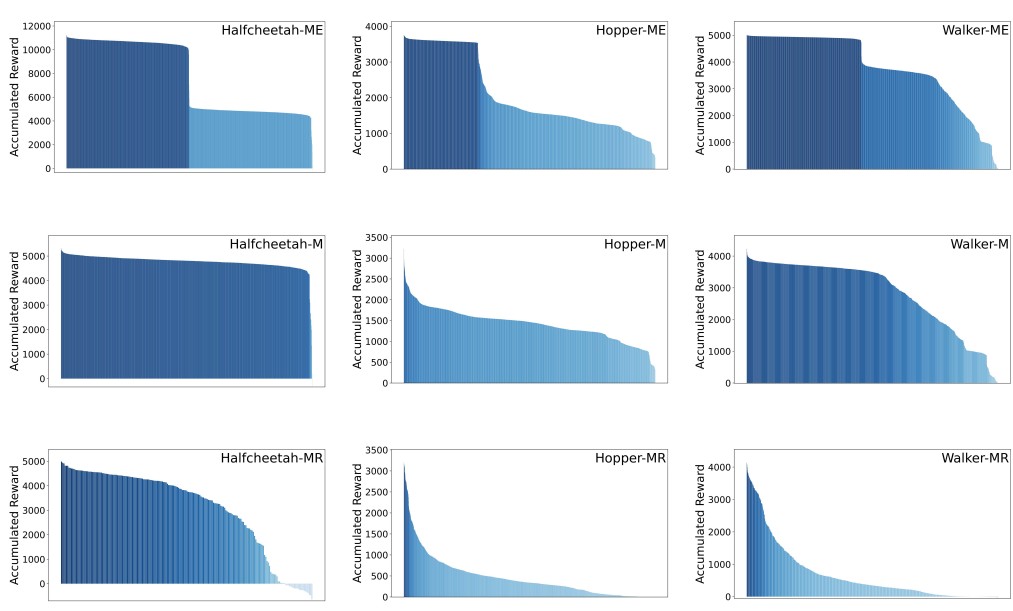

Figure 5: Returns distribution of Med-Expert, Medium and Med-Replay datasets of Halfcheetah, Hopper, Walker2d.

---

**Algorithm 2** Planning and decision-making

---

**Require:** Current state $s_t$, diffusion model $\phi_\theta$, step estimation model $f_\psi$, database $\mathcal{D}$, similarity threshold $\delta$, Top-$k$ selection $k$
1: Retrieve candidate states $\{s_i'\}$ from $\mathcal{D}$ s.t. $\text{sim}(s_t, s_i') \geq \delta$
2: **if** candidates found **then**
3:    Select top-$k$ states by similarity
4:    Re-rank candidates and choose best $s_t^g$ by Eq. 4
5:    Estimate step $\hat{i}_t$ by Eq. 6
6: **end if**
7: Initialize noisy sub-trajectory $\tau_{temp} \sim \mathcal{N}(0, I)$ of length $H \geq \hat{i}_t$
8: Substitute $s_t$ and $s_t^g$ into $\tau_{temp}$ at positions 0 and $\hat{i}_t$
9: **for** $k = K$ down to 1 **do**
10:    Reverse denoise $\tau_{temp}$ using $\phi_\theta$ and guidance $\mathcal{J}_\phi$
11: **end for**
12: Obtain clean trajectory $\hat{\tau}_t^0$
13: **if** trajectory contains state-action pairs **then**
14:    Execute action $\hat{a}_{t+1}$ in environment
15: **else**
16:    Use inverse dynamics: $a_t = f_a(s_t, \hat{s}_{t+1})$ and execute
17: **end if**

---

## D  IMPLEMENTATION DETAILS

- We represent the noise model in D-RAD with a temporal U-Net Janner et al. (2022), consisting of a U-Net structure with 6 repeated residual blocks. Each block consisted of two temporal convolutions, each followed by group norm , and a final Mish nonlinearity. Timestep and condition embeddings, both 128-dimensional vectors, are produced by separate 2-layered MLP (with 256 hidden units and Mish nonlinearity) and are concatenated together before getting added to the activations of the first temporal convolution within each block .

Table 3: Planning horizons and levels used in D-RAD and DL-RAD across different environments.

| Method | Environment | Planning Horizon $H$ | Temporal Jumps / Levels |
|--------|-------------|----------------------|--------------------------|
| D-RAD | MuJoCo (locomotion) | 32 | - |
| D-RAD | Kitchen | 32 | - |
| D-RAD | AntMaze | 64 | - |
| D-RAD | Maze2D U-Maze | 128 | - |
| D-RAD | Maze2D Medium | 265 | - |
| D-RAD | Maze2D Large | 384 | - |
| DL-RAD | Kitchen | 49 | 16, 4, 1 |
| DL-RAD | MuJoCo (locomotion) | 129 | 32, 8, 1 |
| DL-RAD | AntMaze | 129 | 32, 8, 1 |

- For all locomotion tasks, we regard trajectories of length 1000 as expert demonstrations. For AntMaze, expert targets are selected from trajectories whose first hitting step of the goal lies between 150 and 600. For Kitchen, trajectories that successfully complete three designated tasks are considered expert. For Maze2D, expert trajectories are selected according to the number of steps required to reach the goal for the first time: 400-600 steps for Maze2D-medium, 400-800 steps for Maze2D-large, and 200-300 steps for Maze2D-umaze.

- The planning horizons and temporal jumps used in D-RAD and DL-RAD across different environments are summarized in Table 3.

- We consider the top-6 most similar candidates when selecting the target state in D-RAD, and the top-500 most similar candidates when selecting the target state in DL-RAD.

# E  ADDITIONAL EXPERIMENT RESULTS

## E.1  EFFICIENCY

A potential concern with RAD is the inference-time cost from the retrieval component in the TS module, which performs vector-based similarity search and may introduce additional memory and computational overhead.

To characterize the worst-case memory footprint, we measured the cost of storing high-dimensional indices representing the entire state space. This full-state index requires approximately 1.4 GB of CPU memory on our in-house server (502 GB RAM), which remains negligible at system scale. For the same full-state index, a single retrieval including network latency takes about 0.4 seconds, and this cost can be further reduced through local caching.

However, RAD does not query the entire state space during evaluation. All retrieval operations in our actual experiments are performed within a pre-constructed expert database, which is substantially smaller. Consequently, the true time overhead differs from the 0.4-second worst-case measurement above.

To provide an accurate measurement of the actual overhead in RAD, we have measured the total time taken by RAD to generate a single action, including the time for the policy to compute the action from a given state and the time required to execute the action in the environment and transition to the next state. The results are summarized in Table 4.

Table 4: Per-step inference time for different methods.

| Environment | Method | Inference Time (s) | Performance |
|-------------|--------|--------------------|-------------|
| AntMaze | DiffuserLite | 0.06 | 77.6 |
| AntMaze | DL-RAD | 0.26 | 83.3 |
| Locomotion | DiffuserLite | 0.05 | 86.0 |
| Locomotion | DL-RAD | 0.11 | 87.5 |

Compared to DiffuserLite, DL-RAD introduces an additional overhead (approximately +0.20s in AntMaze and +0.06s in Locomotion), mainly due to the retrieval component in the TS module. The observed latency is acceptable for real-time execution in these tasks, given the performance improvement DL-RAD achieves. Moreover, we can further improve the efficiency of retrieval by filtering out suboptimal high-return states.

## E.2 ADDITIONAL ABLATION STUDY

Table 5: Ablation study comparing DL-RAD with and without target states across different environments.

| Environment | No Target States | DL-RAD |
|---|---|---|
| HalfCheetah-MR | 41.9 | 44.4 |
| Hopper-MR | 24.1 | 100.4 |
| Walker2d-MR | 72.4 | 93.5 |

First, to evaluate the contribution of the retrieval module in DL-RAD, we conduct an ablation study in which target states are no longer provided. This setting tests the model's performance when it cannot rely on retrieved target states for guidance. Table 5 summarizes the results across several environments. The version without the retrieval module shows a clear drop in performance compared with the full RAD model.

## E.3 PARAMETER STUDY

Table 6: Effect of minimum similarity threshold $\delta$ for D-RAD.

| $\delta$ | HalfCheetah-M | Hopper-M | Walker2d-M |
|---|---|---|---|
| 0.0 | 43.6 | 54.2 | 64.3 |
| 0.5 | 43.7 | 77.5 | 53.2 |
| 0.8 | 44.0 | 74.8 | 58.4 |
| 0.9 | 44.2 | 82.5 | 82.8 |

Table 7: Effect of minimum similarity threshold $\delta$ for DL-RAD.

| $\delta$ | AntMaze-L-P | Kitchen-M | Maze2d-M | Hopper-MR |
|---|---|---|---|---|
| 0.6 | 60.7 | 0.0 | 52.0 | 100.4 |
| 0.7 | 50.0 | 2.5 | 59.0 | 100.3 |
| 0.8 | 80.0 | 60 | 78.3 | 100.2 |
| 0.9 | 70.0 | 72.7 | 60.1 | 96.5 |

To investigate the effect of the minimum similarity threshold $\delta$ in the target selection module, we conduct experiments varying $\delta$ while keeping other settings fixed. The results are summarized in Table 6 and Table 7.

## E.4 ADDITIONAL DISTRIBUTION SHIFTS EXPERIMENTS

To more systematically evaluate whether the learned policies can generalize to states not present in the training dataset, we conducted additional OOD tests on both Maze2D and AntMaze. For Maze2D, we randomly sampled initial states from maze2d-open-v0 and executed policies trained on maze2d-umaze-v1 or maze2d-medium-v1. For AntMaze, we randomly sampled initial states from antmaze-medium-diverse-v2 and evaluated policies trained on antmaze-medium-play-v2 or antmaze-large-play-v2. The results are reported in Table 8.

Across all datasets, DL-RAD consistently outperforms the baselines, often by a substantial margin. This demonstrates the effectiveness of our method.

Table 8: Performance under distribution shifts.

| Environment | DiffStitch | DiffuserLite | DL-RAD |
|---|---|---|---|
| Antmaze-medium-play | 36.7 | 6.7 | **43.3** |
| Antmaze-large-play | 33.3 | 0.0 | **36.7** |
| Maze2D Medium | 14.7 | 28.3 | **38.0** |
| Maze2D U-maze | 10.8 | 28.5 | **44.7** |

### E.5 SENSITIVITY TO IMPERFECT RANKING IN THE RETRIEVAL MODULE

To examine whether RAD depends heavily on perfect ranking within the retrieval module, we conducted an additional stress test on AntMaze by deliberately degrading the ranking quality. In the final step of the TS module, instead of always selecting the top-1 state, we constructed candidate sets of size 1, 3, 5, and 7, corresponding to increasingly noisy retrieval. For each candidate set, we randomly sampled one state as the retrieved target, thereby simulating scenarios in which the retrieval mechanism returns suboptimal or partially misranked states. The results are summarized in Table 9.

Table 9: Sensitivity to Retrieval Ranking Quality.

| **Environment** | **Top-1** | **Top-3** | **Top-5** | **Top-7** |
|---|---|---|---|---|
| Antmaze-medium-play | 86.7 | 85.3 | 85.3 | 72.0 |
| Antmaze-large-play | 80.0 | 73.3 | 70.0 | 66.7 |
| Antmaze-medium-diverse | 93.3 | 86.7 | 84.3 | 83.3 |
| Antmaze-large-diverse | 73.3 | 62.0 | 58.7 | 62.0 |

As the candidate set grows larger, the noise in the retrieval ranking increases, and the performance shows a gradual downward trend. This behavior is expected: when the retrieved target state is not necessarily the optimal one, the guidance provided to the planner becomes weaker, leading to reduced success rates. More importantly, however, this degradation is gradual rather than catastrophic. Comparing these results against the baselines in Table 1, we observe that even the worst Top-7 performance remains competitive in most environments. For example, in AntMaze-Medium-Diverse, the Top-7 setting still achieves 83.3, ranking among the top three methods.

Therefore, although imperfect ranking introduces some negative effects on the performance, RAD can still benefit from the retrieved target even when it is suboptimal, as long as the retrieved state lies within a reasonably high-value region.

## F PROOF OF ENTROPY REDUCTION WITH TARGET CONDITIONING

To analyze the effect of conditioning on additional target information in trajectory forecasting, we denote the subsequent trajectory generated for planning as a random variable $\tau$, the current state as $s_t$, and the retrieved target state as $s_g$. The predictive uncertainty associated with the trajectory given only $s_t$ is quantified by the conditional entropy $H(\tau \mid s_t)$; larger values indicate greater uncertainty and lower predictive confidence.

When the target state $s_g$ is included as an additional conditioning variable, the uncertainty becomes $H(\tau \mid s_t, s_g)$. By applying the chain rule of conditional entropy to the joint variable pair $(\tau, s_g)$, we obtain two equivalent expressions:

$$H(\tau, s_g \mid s_t) = H(\tau \mid s_t) + H(s_g \mid \tau, s_t) = H(s_g \mid s_t) + H(\tau \mid s_t, s_g).$$

Equating the two decompositions and reorganizing terms gives:

$$H(\tau \mid s_t) - H(\tau \mid s_t, s_g) = H(s_g \mid s_t) - H(s_g \mid \tau, s_t).$$

The right-hand side corresponds to the conditional mutual information $I(\tau; s_g \mid s_t)$, leading to:

$$H(\tau \mid s_t) - H(\tau \mid s_t, s_g) = I(\tau; s_g \mid s_t).$$

Since conditional mutual information is non-negative, i.e.,

$$I(\tau; s_g \mid s_t) \geq 0,$$

we obtain the inequality:
$$H(\tau \mid s_t) \geq H(\tau \mid s_t, s_g).$$

This result demonstrates that conditioning on the retrieved target state preserves or decreases the entropy of the trajectory distribution. Therefore, whenever $I(\tau; s_g \mid s_t) > 0$, the introduction of $s_g$ provides additional relevant information that reduces uncertainty and leads to more accurate and reliable trajectory prediction (i.e.the generation of the subsequent trajectory for planning).

## G COMPARISON WITH TRAJECTORY STITCHING METHODS

For completeness, we provide a detailed discussion on how RAD relates to trajectory-stitching approaches, particularly DiffStitch. RAD is indeed conceptually related to DiffStitch, as both methods are built based on generative models and conduct stitching. However, RAD is different DiffStitch due to:

- DiffStitch is a data augmentation method. It generates a fixed, enlarged data by stitching trajectory segments in the original offline dataset to enhance the offline dataset.
- RAD is an offline RL algorithm. It dynamically retrieves reachable and high-return states as the target states, and uses a diffusion model to plan toward the target states. This enables adaptive high-return-aware planning and decision making.
- Theoretically, the augmented data produced by DiffStitch can be further used to train RAD. This means the two approaches are compatible and can be organically combined to yield more efficient decision-making, rather than being mutually exclusive.

A structured comparison is provided below:

Table 10: Comparison between DiffStitch and RAD.

| Aspect | DiffStitch | RAD |
|---|---|---|
| Type | Diffusion-based data augmentation for offline RL | Diffusion-based offline RL method |
| Stitching | Yes, stitching trajectory segments for data generation | Yes, stitching the current state to the target state for planning |
| Trajectory Planning | N/A | Yes |
| Handling OOD States | Limited(fixed dataset) | Flexible via dynamic retrieval |
| Adaptivity | Static | Dynamic, per-step planning |
| Target | high-return subtrajectory | high-return states |