# OpenReview forum: "RAD: Retrieval High-quality Demonstrations to Enhance Decision-making"
_ICLR.cc/2026/Conference — Submitted to ICLR 2026_

### Official Review · Reviewer_wzXn · 2025-10-15

**Soundness:** 2
**Presentation:** 3
**Contribution:** 2
**Rating:** 4
**Confidence:** 4

**Summary:**

The paper introduces RAD, a retrieval-augmented framework for offline RL that enhances policy generalisation. RAD dynamically retrieves high-return and reachable states from the offline dataset as intermediate targets and employs a diffusion-based generative model to plan sub-trajectories towards them. This retrieval-guided planning allows agents to escape low-return or out-of-distribution regions and achieve higher rewards. Experiments show that RAD consistently matches or outperforms strong baselines.

**Strengths:**

1. The manuscript is well-structured and clearly written.

2. The motivation is strong and effectively highlights the rationale of the method.

**Weaknesses:**

The novelty of this method appears limited. To my understanding, many prior works in this area employ model-based approaches to augment offline datasets, such as various trajectory-stitching methods. The key contribution of this paper seems to lie in combining diffusion models with generalisation beyond the offline dataset. However, diffusion models have already been used for dataset augmentation, and other approaches have separately explored generalisation beyond offline data. As such, this paper mainly integrates these two existing directions, and its conceptual novelty may be insufficient for an ICLR paper.

**Questions:**

1. The authors should cite the related work properly.

2. There are several typos that should be corrected: line 159, line 201 (capital letter), Eq. 13 (subscript), line 294 (definition of $V(s_t)$), and line 406 (an extra line break).

3. I believe that the TS, SE, and PL modules are executed during evaluation, functioning similarly to real-time planning. Therefore, I am concerned about the computational cost during evaluation. Could the authors provide details on the inference time or latency required to produce a single action?

---

> ### Author Response · Authors · 2025-11-22
> **Responses to Reviewer wzXn(Q1-Q3)**
>
> **Q1.** The novelty of this method appears limited. To my understanding, many prior works in this area employ model-based approaches to augment offline datasets, such as various trajectory-stitching methods. The key contribution of this paper seems to lie in combining diffusion models with generalisation beyond the offline dataset. However, diffusion models have already been used for dataset augmentation, and other approaches have separately explored generalisation beyond offline data. As such, this paper mainly integrates these two existing directions, and its conceptual novelty may be insufficient for an ICLR paper.
>
> **A1.** Thank you for your comment. First, we would like to clarify that RAD is a generative-model-based offline RL algorithm rather than a data augmentation method. It improves decision-making by retrieving high-return, reachable states from the offline dataset as targets and using a generative model to plan trajectories toward these targets. To highlight the distinction between our approach and typical generative-model-based data augmentation methods such as DiffStitch, we summarize the comparison results as follows：
>
> | Aspect           | DiffStitch                         | RAD                                         |
> |-------------------------|--------------------------------------------------------------|------------------------------------------------------------------|
> | Type                    | diffusion-based data augmentation method for offline RL      | diffusion-based offline RL method                               |
> | Stitching               | Yes, stitching trajectory segments for data generation       | Yes, stitching the current state to the target state for planning |
> | Trajectory Planning     | N/A                                        | Yes                                                             |
> | Handling OOD States     | Limited (fixed dataset)                                      | Flexible via dynamic retrieval                                  |
> | Adaptivity              | Static                                                       | Dynamic, per-step planning                                      |
> | Target                  | high-return subtrajectory                                    | high-return states                                              |
>
> Therefore, RAD is fundamentally different from generative data augmentation methods, as it focuses on enhancing the decision-making process itself through retrieval-based planning rather than merely expanding the training dataset.
> Second, while some previous works explore generalization beyond offline data, they typically rely on pessimistic value extrapolation, model-based rollouts, or policy constraints. In contrast, RAD provides a distinct mechanism: by retrieving high-return reachable states and planning towards them, it efficiently guides the agent from OOD or low-reward regions to high-return regions in the state space. This closed-loop, retrieval-to-planning approach clearly differentiates RAD from existing generalization strategies.
>
> **Q2.** The manuscript should properly cite the related work and correct several typos and formatting issues.
>
> **A2.** We thank the reviewer for the suggestions and have updated the manuscript to properly cite all relevant related work and to correct all typos and formatting issues in the revised version.
>
> **Q3.** I believe that the TS, SE, and PL modules are executed during evaluation, functioning similarly to real-time planning. Therefore, I am concerned about the computational cost during evaluation. Could the authors provide details on the inference time or latency required to produce a single action?
>
> **A3.** Thank you for the question. Indeed, the TS, SE, and PL modules are executed during evaluation. Despite this, the overall per-step inference time remains manageable. Specifically, we have measured the total time taken by RAD to generate a single action, including the time for the policy to compute the action from a given state and the time required to execute the action in the environment and transition to the next state. The result summarized below:
>
> | Environment | Method        | Inference Time (s) | Performance |
> |------------|---------------|------------------|------------|
> | AntMaze    | DiffuserLite  | 0.06             |      77.6      |
> | AntMaze    | DL-RAD        | 0.26             |       83.3     |
> | Locomotion | DiffuserLite  | 0.05             |        86.0    |
> | Locomotion | DL-RAD        | 0.11             |   87.5         |
>
> Compared to DiffuserLite, DL-RAD introduces an additional overhead (approximately +0.20s in AntMaze and +0.06s in Locomotion), mainly due to the retrieval component in the TS module. The observed latency is acceptable for real-time execution in these tasks, given the performance improvement DL-RAD achieves. Moreover, we can further improve the efficiency of retrieval by filtering out suboptimal high-return states.

---

> ### Author Response · Authors · 2025-11-25
> **Do Our Responses Resolve Your Main Concerns?**
>
> Dear Reviewer wzXn,
>
> I hope this message finds you well. We have provided detailed responses to your insightful comments, which we hope address them satisfactorily. Some of the key clarifications include the following:
>
> 1. **Clarification on novelty**. RAD is a generative-model-based offline RL method that retrieves high-return states as planning targets, improving decision-making and guiding the policy from OOD or low-reward regions, unlike pessimistic value or rollout-based methods.
>
> 2. **Inference computational cost**. We report single-step action generation time for RAD and baselines. Considering RAD’s performance improvements, the observed latency is acceptable for real-time execution.
>
> We sincerely hope that these clarifications have resolved your concerns.  If so, we would greatly appreciate it if you could consider updating your evaluation score in light of these clarifications. If there are any additional points or feedback you would like us to consider, please let us know. Your insights are invaluable to us, and we are eager to address any remaining issues to improve our work.
>
> Thank you for your time and effort in reviewing our paper.
>
> Best regards,
>
> The Authors

---

> > ### Comment · Reviewer_wzXn · 2025-11-25
> >
> > Thank you for your detailed explanation. After giving it careful thought, I’m sorry that I would like to maintain my score.

---

> > > ### Author Response · Authors · 2025-11-25
> > >
> > > Dear Reviewer wzXn,
> > >
> > > Thank you for your time and for reconsidering our responses. We fully respect your decision to maintain your score.
> > >
> > > To help us better understand and further improve the work, could you kindly let us know which concerns you feel remain insufficiently addressed? We would be very happy to provide additional clarification or engage in further discussion on any specific points that are still unclear.
> > >
> > > Your feedback is extremely valuable to us, and we greatly appreciate your willingness to communicate.
> > >
> > > Best regards,
> > >
> > > The Authors

---

### Official Review · Reviewer_woTf · 2025-10-30

**Soundness:** 3
**Presentation:** 2
**Contribution:** 3
**Rating:** 2
**Confidence:** 4

**Summary:**

This paper presents a method for improving offline RL. Their method (RAD) focuses on three components, (1) a target picker that from a state finds a reachable state with high return, (2) a step estimator to estimate how far the target position is from the current position and (3) a planner that creates a trajectory to get to the goal states. The high level idea is to train these components and use them to act in the environment. They perform experiments on offline RL tasks to compare to existing baselines and ablate the components of their method. They find that their method is able to perform as well as or better as baselines on their experiments and that their components all contribute to their final results. They also perform experiments on the generalization of their method and find it is able to reasonably generalize.

**Strengths:**

This work has reasonable novelty. This work isn't directly proposing a diffusion model but instead builds upon other works models and adds in a method for guiding these models to create the training data.

The experimental evidence is reasonable. They compare to many baselines on a reasonable set of experiments. I would really like to see 95% confidence intervals here (like in table 1) as without them it is harder to distinguish great results from good ones.

The actual results on the tasks perform quite well. In general this method is doing much better than the baselines.

**Weaknesses:**

In general I feel like the writing could be more clear and I think we are missing some information that I feel is critical. I will put my questions in the question section but I don't feel confident I understand how your method actually runs. Paragraph near 155 - This paragraph is pretty hard to read/understand, what does "transit" mean here? Do you mean a trajectory from s_t to s_t^g?.

The limitations of this method are not properly addressed. I'll put specific questions in the question part again but some limitations that I can think of. Speed - running diffusion models and querying a large dataset seems slow. Long horizon - there is a limit to your planner right so does increasing this limit cause issues with attempting to plan from your proposed state?

In general I come away from the paper with a lot of questions. I'll put them below but if you can answer them and update the paper then the weaknesses will be minimal and I will gladly raise my score to an accept since this is an interesting paper but I just don't fully understand the method and weaknesses.

**Questions:**

How do you actually run your policy? Do you run one step of your method, then take the action or do you create a trajectory with your planner and run it open loop? How long does this take? Is it a feasible real-time method or not?

You say that you can just follow high return actions once you get to the in distribution states but how are these actions computed? Is it still from your method? Or are we following the exact actions in the dataset? Is the method used to simply get out of OOD situations and then we run a normal policy?

Diffusion models are slow, how long does your method take with the extra search on top? If you have to query your entire state space when you run your method how long does that take? How would this deal with a large amount of offline data?

212 - So the idea is you want to find the number of steps to a new state but doesn't the feature vector contain the position in each trajectory the state is at? Is there a problem with assuming we are in the "same" location as long as the difference between states is small enough?

Small thing but make sure to check for typos as well there are a handful.

---

> ### Author Response · Authors · 2025-11-22
> **Responses to Reviewer woTf(Q1-Q3)**
>
> **Q1.** Clarify the meaning of 'transit' in the paragraph containing line 155; is it referring to a trajectory from $s_t$ to $s_t^g$.
>
> **A1.**  Yes, in the paragraph containing line 155, “transit” refers to a multi-step (non-adjacent) transition from current state $s_t$ to a target state  $s_{t}^g$, where$s_{t}^g = s_{t+k}$ (with $k > 1$). Here, $s_{t}^g$can be viewed as the state that the agent aims to reach after several decision steps. The intuition is that if $s_t^g$ corresponds to a state that is more likely to yield high returns, then guiding the agent toward $s_t^g$ increases the likelihood that it will select actions conducive to achieving higher rewards.
>
> **Q2.** How do you actually run your policy? Do you run one step of your method, then take the action or do you create a trajectory with your planner and run it open loop? How long does this take? Is it a feasible real-time method or not?
>
> **A2.** We thank the reviewer for raising this important question. In our framework, the policy operates in a step-by-step, closed-loop manner. Specifically, given the current state $s_t$, our method proceeds as follows:
>
> (1) The Target Selection (TS) module retrieves a target state  $s_t^g$;
>
> (2) The Step Estimation (SE) module predicts the number of steps  $\hat{i}$ required to reach the target according to $s_t$ and $s_t^g$;
>
> (3) The Planning (PL) module is invoked to generate a sub-trajectory conditioned on $(s_t, s_t^g, \hat{i})$.
>
> (4) An action is then obtained from the generated sub-trajectory and executed in the environment, producing the next state $s_{t+1}$, after which the loop repeats from step (1).
>
> The process could be summarized as follows:
>
>
>   Algorithm: RAD Action Selection (per environment step)
>
> Input: current state $s_t$
>
> while not done:
>
>   &nbsp;&nbsp;  $s_t^g \gets \text{TS}(s_t)$  &nbsp;&nbsp;# target retrieval
>   &nbsp;&nbsp;  $\hat{i} \gets \text{SE}(s_t, s_t^g)$  &nbsp;&nbsp;# step prediction
>   &nbsp;&nbsp;  $\hat{\tau} \gets \text{PL}(s_t, s_t^g, \hat{i})$  &nbsp;&nbsp;# trajectory/state sequence generation
>
>   &nbsp;&nbsp; if D-RAD then
>   &nbsp;&nbsp;&nbsp;&nbsp; $a_t \gets \hat{\tau}[0].\text{action}$  &nbsp;&nbsp;# directly use action from trajectory
>   &nbsp;&nbsp; else DL-RAD
>   &nbsp;&nbsp;&nbsp;&nbsp;  $a_t \gets \text{InverseDynamics}(\hat{\tau}[0].\text{state}, \hat{\tau}[1].\text{state})$
>   &nbsp;&nbsp; end if
>
>   &nbsp;&nbsp;  $s_{t+1} \gets \text{Transition}(s_t, a_t)$  &nbsp;&nbsp;# interact with environment
>   &nbsp;&nbsp;  $s_t \gets s_{t+1}$
>
>  end while
>
> As for the inference time, we have measured the total time taken by RAD to generate a single action, including the time for the policy to compute the action from a given state and the time required to execute the action in the environment and transition to the next state. All inference experiments were conducted on a machine equipped with an Intel Gold 5220 CPU and an NVIDIA A40 GPU. Under this setup, DL-RAD requires approximately 0.26s in AntMaze (about +0.20s compared to DiffuserLite) and 0.11s in Locomotion (about +0.06s compared to DiffuserLite). The observed latency is acceptable for real-time execution in these tasks. Given the performance improvement of DL-RAD over DiffuserLite, we believe that this additional latency is justified.
>
>
> **Q3.** You say that you can just follow high return actions once you get to the in distribution states but how are these actions computed? Is it still from your method? Or are we following the exact actions in the dataset? Is the method used to simply get out of OOD situations and then we run a normal policy?
>
> **A3.** We would like to clarify that “following high-return actions in in-distribution states” refers to leveraging the decision patterns associated with high-return behaviors in the offline dataset, rather than directly executing the raw actions stored in the dataset. At test time, the actions are always produced by our learned policy network, which has already captured these high-return behavioral patterns during training. Therefore, The retrieval mechanism is not only used to mitigate out-of-distribution scenarios, but more importantly, it provides additional guidance that enables the model to make more effective decisions.
>
> Regarding the reviewer’s question about whether our method only helps the agent escape OOD regions and then switches to a “normal policy”: the answer is no. Our method does not merely return the agent to in-distribution states and then stop being used. Instead, the same policy learned with our method is used throughout the entire decision-making process, both in OOD and in-distribution regions. The method is therefore not a two-stage procedure but a unified learning framework.

---

> ### Author Response · Authors · 2025-11-22
> **Responses to Reviewer woTf(Q4-Q6)**
>
> **Q4.** Diffusion models are slow, how long does your method take with the extra search on top? If you have to query your entire state space when you run your method how long does that take? How would this deal with a large amount of offline data?
>
> **A4.**
> (1). We appreciate the reviewer's comment regarding computational efficiency. We have measured the total time taken by RAD to generate a single action, including the time for the policy to compute the action from a given state and the time required to execute the action in the environment and transition to the next state. In the Antmaze environment, DL-RAD takes about 0.26s (about 0.20 seconds longer than DiffuserLite), while in the Locomotion environment, it takes approximately 0.11s (approximately 0.06 seconds longer than DiffuserLite).  The observed latency is acceptable for real-time execution in these tasks, given the performance improvement DL-RAD achieves. Moreover, retrieval occurs only during the inference phase, meaning the added computational overhead is negligible compared to the training cost.
>
> (2) Regarding the query cost over the state space: querying the full state space would take approximately 0.4 seconds per retrieval in locomotion tasks (including network latency). However, RAD does not search the entire state space. Instead, retrieval is performed over a pre-constructed expert database, significantly reducing the lookup cost. As noted in point (1), in practice, generating a complete action sequence takes only 0.11 seconds, which is considerably smaller than the cost required to search the full state space.
>
> (3). To handle large-scale offline data efficiently, we employ two main mechanisms to accelerate retrieval:
>
> a. Constructing a specialized expert experience database, and
>
> b. Limiting the number of returned results by retrieving only the top-$K$ nearest neighbors.
>
>
> **Q5.** So the idea is you want to find the number of steps to a new state but doesn't the feature vector contain the position in each trajectory the state is at? Is there a problem with assuming we are in the "same" location as long as the difference between states is small enough?
>
> **A5.** Thank you for your question.
>
> (1). First of all, we would like to clarify that the core contribution of our work is not merely to estimate the step count between two states.  Rather, the central idea of RAD is to introduce a retrieval mechanism into the planning process to assist the model in making more effective decisions. By retrieving high-return states as planning targets, RAD alleviates the model's poor performance in out-of-distribution (OOD) and sparse-reward scenarios.
>
> (2). The Step Estimation (SE) module is indispensable in our framework. Our method generates trajectory for planning based on a diffusion model which relies not only on state information but also on step count information to generate sub-trajectories. Specifically, given the current state $s_t$, the target state $s_t^g$, and the estimated step count $\hat{i}$, the PL module initializes a noisy sub-trajectory $\tau_{temp}$ of length  $H(≥\hat{i})$, then substitutes ${s}_t$ and ${s}_t^g$ into their respective positions. The reverse denoising process of the diffusion model is then applied to generate a clean sub-trajectory. This means the step count directly determines the temporal index of the target state in the trajectory, ensuring that the generated trajectory is temporally consistent. While the feature vector does encode a state’s position within a trajectory, the step count provides explicit and reliable temporal guidance, which is crucial for accurately generating sub-trajectories. As shown in Figure 3, removing the SE module significantly degrades performance, further highlighting its importance.
>
> (3). As for the issue of assuming two states are the same when their difference is small, we understand your question as asking whether two states could assume being in the “same” location  **relative to the target state** as long as the difference between states is small enough. This is a correct observation. In continuous control environments, the dynamics are locally continuous: states that are highly similar tend to produce similar short-term transitions when executing the same actions.
>
> **Q6.** Small thing but make sure to check for typos as well there are a handful.
>
> **A6.** We thank the reviewer for pointing this out and have carefully corrected all typos in the revised version.

---

> ### Author Response · Authors · 2025-11-25
> **Do Our Responses Resolve Your Main Concerns?**
>
> Dear Reviewer woTf,
>
> I hope this message finds you well. We have provided detailed responses to your insightful comments, which we hope address them satisfactorily. Some of the key clarifications include the following:
>
> 1. **Policy execution**. RAD runs in a closed-loop manner: TS selects a target, SE predicts steps, PL generates a sub-trajectory, and an action is executed.
> 2. **Execution time and data handling**. Single-step latency is acceptable given RAD’s performance, and retrieval scales efficiently to large offline datasets.
> 3. **Core contribution and importance of the SE module**. We clarify the core contribution of our work and explain the importance of the SE module.
>
> We sincerely hope that these clarifications have resolved your concerns.  If so, we would greatly appreciate it if you could consider updating your evaluation score in light of these clarifications. If there are any additional points or feedback you would like us to consider, please let us know. Your insights are invaluable to us, and we are eager to address any remaining issues to improve our work.
>
> Thank you for your time and effort in reviewing our paper.
>
> Best regards,
>
> The Authors

---

### Official Review · Reviewer_6yLX · 2025-10-31

**Soundness:** 2
**Presentation:** 2
**Contribution:** 3
**Rating:** 4
**Confidence:** 3

**Summary:**

The paper proposes RAD (Retrieval High-quality Demonstrations), a retrieval-augmented trajectory stitching method for offline RL, which is composed of three modules as Target Selection (TS), Step Estimation(ES), and Planning (PL). Given a current state, RAD (i) retrieves similar, high-return, and purportedly reachable target states from an offline dataset; (ii) estimates the temporal distance (step span) to the target; and (iii) conditions a diffusion planner (Diffuser or DiffuserLite) to generate a sub-trajectory that steers the agent toward the target, after which the policy continues along high-return actions in the demonstration. Experiments on D4RL tasks report broadly competitive results, plus ablations on each module and a generalization test on Random datasets.

**Strengths:**

The proposed RAD method combines target state retrieval with diffusion-based planning in offline RL. While trajectory stitching and diffusion planners exist, RAD’s idea of adaptive target retrieval at inference time (instead of static, offline augmentation) is a meaningful design point, appealing in sparse-reward or long-horizon domains (e.g., AntMaze), where “latching onto” good sub-goals helps escape low-value regions.

The “target-then-plan” decomposition (TS → ES → PL) is conceptually clear, and may be applied to many sequence-modeling or model-free offline baselines. The presentation of three modules and the illustrations make the paper easy to follow.

The experiment done in this paper is quite comprehensive, including a solid suite of baselines, visualization, ablations, and a small distribution-shift study.

**Weaknesses:**

The idea is straightforward, but lacks theoretical support. Firstly, “reachability” is asserted, not guaranteed. TS currently filters by cosine similarity and high return, then picks the candidate with the longest remaining length; there is no principled guarantee that the target is reachable without collisions/obstacles under the learned dynamics—especially salient in mazes or any environment with an inconsistent transition model (e.g., a wall separating the two near states exists and has not been explored much in the dataset). The author trained an extra binary classifier to predict the connectability in Appendix D for D-RAD, but did not show that in the main text, nor did extra comparison on that. Besides, the pipeline seems deterministic once top-k are retrieved (tie-breaks aside). This can induce policy shift brittleness: if the top candidate is slightly wrong or not reachable, the planner may commit to a poor target with no alternative selection. For potential improvement, maybe introduce stochastic target sampling. Finally, the way to select high returns also seems problematic. The cumulative discounted reward from step t is defined as $v_t = \sum_{i \ge t } \gamma^i r_i$ instead of $v_t = \sum_{i \ge t } \gamma^{i-t}r_i$, which means that earlier states will always have a higher return in an environment with non-negative rewards.

Experiment Issues: Only 3 seeds and no standard deviations/confidence intervals. The table highlights entries within 0.95×MAX, which is an unusual criterion and can obscure variance. Besides, it will be clearer if an average score is presented.

The proposed RAD seems closely related to trajectory stitching-based methods like DiffStitch, but there is no further comparison and discussion in the related works.

Minor issues:

- Typos (e.g., Line 159 “makinga”; Line 157 “TS then estimates the step” should be ES; Inconsistent capitalization for “Diffuserlite”; “Diverde” in Table 1).

- Notation Inconsistency (‘G’ is used both as the goal state in Introduction and the return in 4.1. The latter could be replaced as $R(\tau)$ denoted in equation (2); $f_e$ refers to different things in 4.2 and Appendix D)

- 4.3 presents two sub-trajectory types (noisy state-action pair vs. noisy state), but no context for which one is used.

**Questions:**

1. Please clarify or correct the definitions of v_t in 3.2 and anywhere else they propagate.

2. Is ES trained with cross-entropy over H–1 classes as suggested in 4.2 or with the MSE in Eq. (14)? If the latter, how do you backprop through argmax?

3. Can you show ablation results on how much the classifier mentioned in Appendix D helps to prevent TS from selecting an unreachable target state?

4. Why does DL-RAD underperform in Maze2D compared with other baselines?

---

> ### Author Response · Authors · 2025-11-22
> **Responses to Reviewer 6yLX(Q1-Q2)**
>
> **Q1.** The idea is straightforward, but lacks theoretical support.
>
> **A1.** Thank you for your question.
>
> We would like to briefly recap the method before presenting the theoretical analysis.
> First of all,  RAD aims to address the challenges of offline RL, where a finite static dataset fundamentally limits generalization. In situations where the agent encounters states outside the training distribution or regions with sparse rewards,  it becomes inherently difficult to make reliable decisions. A common approach in offline RL is to leverage a learned generative model to generate the subsequent trajectories for planning. However, planning in OOD regions often suffers from substantial uncertainty due to model errors and distributional shift, making effective generation challenging.
> RAD reduces this uncertainty by retrieving a high-return state as a planning target, providing an additional observation that serves as an anchor to constrain the generative planning process and guide the agent toward more reliable decision-making.
>
> Second, we present a theoretical analysis to justify the method. We denote the subsequent trajectory generated for as a random variable  $\tau$ , the current state as  $s_t$, and the retrieved target state as  $s_g$ . The predictive uncertainty of the trajectory given the current state is quantified by the conditional entropy $H(\tau \mid s_t)$, where a higher entropy reflects a higher degree of uncertainty and thus lower predictive reliability. When the goal information  $s_g$  is incorporated as an additional conditioning variable, the uncertainty becomes  $H(\tau \mid s_t, s_g)$. Applying the chain rule of conditional entropy to the joint variable pair $(\tau, s_g)$ yields two equivalent decompositions:  $H(\tau, s_g \mid s_t) = H(\tau \mid s_t) + H(s_g \mid \tau, s_t) = H(s_g \mid s_t) + H(\tau \mid s_t, s_g)$. Rearranging terms leads to  $H(\tau \mid s_t) - H(\tau \mid s_t, s_g) = H(s_g \mid s_t) - H(s_g \mid \tau, s_t)$, which matches the definition of conditional mutual information   $I(\tau; s_g \mid s_t)$. Since conditional mutual information is always non-negative, i.e.,   $I(\tau; s_g \mid s_t) \ge 0$ , it follows that   $H(\tau \mid s_t) \ge H(\tau \mid s_t, s_g)$. This result demonstrates that conditioning on the retrieved target state cannot increase prediction uncertainty; instead, it either preserves or decreases the entropy of the trajectory distribution. Therefore, whenever  $I(\tau; s_g \mid s_t) > 0$, the introduction of  $s_g$  provides additional relevant information that reduces uncertainty and leads to more accurate and reliable trajectory prediction (i.e.the generation of the subsequent trajectory for planning).
>
>
> **Q2.** "Reachability" is asserted, not guaranteed. Since TS currently filters by cosine similarity and high return, then picks the candidate with the longest remaining length, how can we be sure that the target is actually reachable without collisions or obstacles under the learned dynamics (especially in mazes or environments with inconsistent transition models)?
>
> **A2.** We appreciate the reviewer’s insightful question. Indeed, as noted, target “reachability” cannot be guaranteed in theory. However, our method is designed to make the retrieved target states more likely to be reachable in practice.
>
> In continuous control environments, we exploit the local continuity of the dynamics, i.e., states that are close in the state space tend to have similar short-term evolution under the same actions. Concretely, our Target Selection module retrieves candidate states that are (1) semantically similar to the current state $𝑠_𝑡$ and (2) part of high-return trajectories. By further prioritizing candidates with longer remaining trajectory length, we bias the selection toward targets that are reachable via sequences of successful actions observed in the offline dataset. It ensures that the chosen targets are highly likely to be reachable under the learned dynamics.
>
> Empirically, this approach proves effective. As shown in Table 1, RAD achieves top or near-top performance on 16 out of 18 datasets. Notably, in the Antmaze environments—where transition consistency is critical—DL-RAD attains an average score of 83.3, outperforming the second-best method (DiffuserLite) by 5.7 points, demonstrating its robustness even in challenging settings with complex dynamics.

---

> ### Author Response · Authors · 2025-11-22
> **Responses to Reviewer 6yLX(Q3-Q6)**
>
> **Q3.** The pipeline seems deterministic once top-k are retrieved (tie-breaks aside). Could this induce policy shift brittleness — for example, if the top candidate is slightly wrong or not reachable, might the planner commit to a poor target with no alternative selection?
>
> **A3.** You have raised a valid concern. Indeed, if the top candidate target is slightly inaccurate or unreachable, the planner could potentially commit to a suboptimal goal.
> However, such cases have minimal impact on our method. First, RAD already incorporates the following safeguards to mitigate this risk:
>
> (1) Candidate targets are filtered via cosine similarity in the state space for semantic and geometric consistency.
>
> (2) Candidate targets are then ranked using true return-to-go values from the dataset, avoiding biases from learned value estimators.
>
> (3) Only targets belonging to long, continuous trajectories are retained.
>
> (4) If no high-quality target passes the filters, the model defaults to short-horizon planning via the diffusion model.
>
> (5) Closed-loop execution at each step enables continuous correction of potential target errors.
>
> Moreover, while we retrieve a top-$k$ set of candidates to improve target selection robustness, only one final target state is actually used for decision-making. This means that as long as at least one state in the dataset satisfies our filtering criteria, a valid target can be obtained—significantly reducing the difficulty of acquiring a suitable target.
>
> To understand whether selecting the suboptimal goal would harm performance, we conducted an additional study on AntMaze by intentionally degrading the ranking quality. Specifically, in the final step of the TS module, instead of always selecting the top-1 state, we formed candidate sets of size 1, 3, 5, and 7 and randomly picked one target from each set. That indicates the target is not always the optimal one when the size of the candidate set is 3,5,7. The results are shown below:
>
> |Environment|Top-1|Top-3|Top-5|Top-7|
> |-|:-:|:-:|:-:|:-:|
> |Antmaze-medium-play|$86.7$|$85.3$|$85.3$|$72.0$|
> |Antmaze-large-play|$80.0$|$73.3$|$70.0$|$66.7$|
> |Antmaze-medium-diverse|$93.3$|$86.7$|$84.3$|$83.3$|
> |Antmaze-large-diverse|$73.3$|$62.0$|$58.7$|$62.0$|
>
> As the candidate set grows larger, the noise in the retrieval ranking increases, and the performance shows a gradual downward trend. This behavior is expected: when the retrieved target state is not necessarily the optimal one, the guidance provided to the planner becomes weaker, leading to reduced success rates. However, comparing these results against the baselines in Table 1 of the paper, we observe that even the worst Top-7 performance remains competitive. For example, in AntMaze-medium-diverse, the Top-7 setting still achieves 83.3, ranking among the top three methods. Therefore, although imperfect ranking introduces some negative effects on the performance, RAD can still benefit from the retrieved target even when it is suboptimal, as long as the retrieved state lies within a reasonably high-value region.
>
> **Q4.** Please clarify or correct the definitions of $v_t$ in 3.2 and anywhere else they propagate.
>
> **A4.** We thank the reviewer for pointing this out. We correct this in the revised manuscript to avoid confusion. Importantly, all results and target selection in our experiments are based on this correct formula.
>
>
>
> **Q5.** Could the authors include variance or uncertainty information?
>
> **A5.** We thank the reviewer for the suggestion. We have updated Table 1 in the revised manuscript to include the requested information.
>
> **Q6.** The table highlights entries within $0.95 \times \text{MAX}$, which is an unusual criterion and can obscure variance. Would it be clearer to present average scores instead?
>
> **A6.** We thank the reviewer for the suggestion. The average scores have been added in the revised Table 1. The newly added average scores provide a clearer view of overall performance. Using a $0.95 \times \text{MAX}$ criterion to highlight top-performing entries follows common practice in prior offline RL works[1,2,3].
>
> [1]. Kostrikov, I., Nair, A., and Levine, S. Offline reinforcement learning with implicit Q-learning. In International Conference on Learning Representations, 2022.
>
> [2]. Janner M, Du Y, Tenenbaum J B, et al. Planning with diffusion for flexible behavior synthesis[J]. arXiv preprint arXiv:2205.09991, 2022.
>
> [3]. Dong Z, Hao J, Yuan Y, et al. Diffuserlite: Towards real-time diffusion planning[J]. Advances in Neural Information Processing Systems, 2024, 37: 122556-122583.

---

> ### Author Response · Authors · 2025-11-22
> **Responses to Reviewer 6yLX(Q7-Q11)**
>
> **Q7.** How does RAD relate to trajectory-stitching methods like DiffStitch? Please discuss or compare.
>
> **A7.** We thank the reviewer for raising this relevant comparison. RAD is indeed conceptually related to DiffStitch, as both methods are built based on generative models and conduct stitching. However, RAD is different DiffStitch due to:
>
> (1). DiffStitch is a data augmentation method. It generates a fixed, enlarged data by stitching trajectory segments in the original offline dataset to enhance the offline dataset.
>
> (2). RAD is an offline RL algorithm. It dynamically retrieves reachable and high-return states as the target states, and uses a diffusion model to plan toward the target states. This enables adaptive high-return-aware planning and decision making.
>
> (3). Theoretically, the augmented data produced by DiffStitch can be further used to train RAD. This means the two approaches are compatible and can be organically combined to yield more efficient decision-making, rather than being mutually exclusive.
>
> To further clarify the differences, we summarize them below:
>
> | Aspect                  | DiffStitch                                                   | RAD                                                             |
> |-------------------------|--------------------------------------------------------------|------------------------------------------------------------------|
> | Type                    | diffusion-based data augmentation method for offline RL      | diffusion-based offline RL method                               |
> | Stitching               | Yes, stitching trajectory segments for data generation       | Yes, stitching the current state to the target state for planning |
> | Trajectory Planning     | N/A                                                          | Yes                                                             |
> | Handling OOD States     | Limited (fixed dataset)                                      | Flexible via dynamic retrieval                                  |
> | Adaptivity              | Static                                                       | Dynamic, per-step planning                                      |
> | Target                  | high-return subtrajectory                                    | high-return states                                              |
>
>
> **Q8.** Typos and notation inconsistencies .
>
> **A8.** We thank the reviewer for these careful observations. All mentioned typos and notation inconsistencies have been corrected in the revised manuscript.
>
> **Q9.** Section 4.3 presents two sub-trajectory types (noisy state-action pair vs. noisy state), but gives no context for which one is used. Could the authors clarify this?
>
> **A9.** We thank the reviewer for the comment. Both sub-trajectory types are used in our work: noisy state-action pairs are used to generate the subsequent trajectory for decision making in D-RAD, while the noisy states are used to generate the subsequent state sequence for decision making. We have added the corresponding clarification in the manuscript to avoid ambiguity.
>
> **Q10.** There appear to be inconsistencies between the description of the connectability classifier in Appendix D and the main text, and between the loss function for the Step Estimation module (cross-entropy vs. the MSE form shown in Eq. 14).
>
> **A10.** We thank the reviewer for pointing this out. Both discrepancies came from earlier implementation drafts that were mistakenly left in the submitted version. The connectability classifier mentioned in Appendix D is not part of the final method and has been removed. The Step Estimation module is trained using cross-entropy classification, and the manuscript has been updated accordingly. All descriptions are now consistent with the final implementation.
>
> **Q11.** Why does DL-RAD underperform in Maze2D compared with other baselines?
>
> **A11.**  Thank you for your comment. DL-RAD is built upon DiffuserLite. As shown in Table 1, DiffuserLite’s performance on Maze2D is relatively limited, likely due to the lack of prior evaluation and tuning on this type of sparse-reward, navigation-focused environment. Since DL-RAD uses DiffuserLite as its backbone, its performance on Maze2D is also constrained by these factors. Nevertheless, DL-RAD consistently improves over DiffuserLite, highlighting the benefit of the retrieval-based guidance mechanism in improving trajectory generation and policy performance.

---

> ### Author Response · Authors · 2025-11-25
> **Do Our Responses Resolve Your Main Concerns?**
>
> Dear Reviewer 6yLX,
>
> I hope this message finds you well. We have provided detailed responses to your insightful comments, which we hope address them satisfactorily. Some of the key clarifications include the following:
>
> 1. **Theoretical justification**. We supplemented the theoretical analysis to justify the validity of the proposed method.
> 2. **Mechanism explanation**. We clarified how the retrieval-guided planning paradigm effectively increases the reachability of the retrieved target state.
> 3. **Robustness**. We tested the method’s safeguards against slightly inaccurate or unreachable targets by degrading ranking quality. Performance decreases gradually with noise but remains competitively strong.
> 4. **Method positioning and related work**. We further clarified the differences and connections between RAD and trajectory-stitching methods.
>
> We sincerely hope that these clarifications have resolved your concerns.  If so, we would greatly appreciate it if you could consider updating your evaluation score in light of these clarifications. If there are any additional points or feedback you would like us to consider, please let us know. Your insights are invaluable to us, and we are eager to address any remaining issues to improve our work.
>
> Thank you for your time and effort in reviewing our paper.
>
> Best regards,
>
> The Authors

---

### Official Review · Reviewer_Wzph · 2025-11-01

**Soundness:** 3
**Presentation:** 3
**Contribution:** 3
**Rating:** 6
**Confidence:** 3

**Summary:**

This paper proposes RAD (Retrieval High-quality Demonstrations), a retrieval-augmented offline reinforcement learning framework. Instead of relying on static data augmentation, RAD dynamically retrieves high-return and reachable states from an offline dataset as intermediate targets. It then employs a diffusion-based generative model (built upon Diffuser or DiffuserLite) to generate sub-trajectories toward these targets, improving generalization beyond the dataset distribution. Experiments on D4RL benchmarks show consistent or superior performance over baselines across locomotion, navigation, and manipulation tasks.

**Strengths:**

1. Integrating state retrieval into offline RL is conceptually appealing, and the paper provides a clear motivation for using retrieved high-return states as adaptive guidance for policy improvement.
2. The experiments cover a wide range of D4RL tasks (MuJoCo, AntMaze, Kitchen, Maze2D) with solid baselines including model-free, model-based, and diffusion-based methods. RAD demonstrates competitive or superior performance on most datasets.

**Weaknesses:**

1. Several sections contain minor grammatical errors and redundant phrasing (e.g., “novelly integrates,” “makinga decision”). Figures could be improved for clarity and caption detail.
2. The distribution-shift test (training on Medium-Replay, testing with Random starts) is limited to three environments; more systematic OOD tests would strengthen claims.
3. Although the retrieval mechanism is new, the overall architecture largely reuses existing components from Diffuser/DiffuserLite, and the retrieval is applied at the state level without deep theoretical justification for its optimality or stability.

**Questions:**

1. How sensitive is RAD to the accuracy of the value-based ranking in target retrieval? Would errors in return estimation significantly degrade performance?
2. Can RAD handle multi-modal retrieval results (e.g., when several high-return trajectories exist but lead to different goals)?

---

> ### Author Response · Authors · 2025-11-22
> **Responses to Reviewer Wzph(Q1-Q3)**
>
> **Q1.** Minor grammatical errors and suboptimal figure clarity.
>
> **A1.** We thank the reviewer for the feedback and have revised the manuscript.
>
> **Q2.** The distribution-shift test (training on Medium-Replay, testing with Random starts) is limited to three environments; more systematic OOD tests would strengthen claims.
>
> **A2.** We thank the reviewer for the suggestion.
> To more systematically evaluate whether the learned policies can generalize to states not present in the training dataset, we conducted additional OOD tests on both Maze2D and AntMaze. For AntMaze, we randomly sampled initial states from antmaze-medium-diverse-v2 and evaluated policies trained on antmaze-medium-play-v2 or antmaze-large-play-v2. For Maze2D, we randomly sampled initial states from maze2d-open-v0 and executed policies trained on maze2d-umaze-v1 or maze2d-medium-v1. The results are shown below:
>
> |Environment|DiffStitch|DiffuserLite|DL-RAD|
> |-|:-:|:-:|:-:|
> |Antmaze-medium-play|$36.7$|$6.7$|$43.3$|
> |Antmaze-large-play|$33.3$|$0.0$|$36.7$|
> |Maze2D Medium|$14.7$|$28.3$|$38.0$|
> |Maze2D U-maze|$10.8$|$28.5$|$44.7$|
>
> Across all datasets, DL-RAD consistently outperforms the baselines, often by a substantial margin. This demonstrates the effectiveness of our method.
>
> **Q3.** Although the retrieval mechanism is new, the overall architecture largely reuses existing components from Diffuser/DiffuserLite, and the retrieval is applied at the state level without deep theoretical justification for its optimality or stability.
>
> **A3.** We thank the reviewer for the insightful feedback regarding architectural reuse and theoretical motivation.
>
> (1). For the architectural reuse, although our architecture reuses components from Diffuser and DiffuserLite, our contribution differs fundamentally: Instead of designing an entirely new offline RL algorithm, we focus on introducing a simple yet effective retrieval mechanism on planning-based offline RL. This mechanism transforms conventional planning (typically unaware of high-return trajectories) into a high-return-aware and target-oriented process. The proposed retrieval-guided paradigm is model-agnostic and can be seamlessly integrated with any generative-model-based planner, enabling more effective decision-making in both out-of-distribution and sparse-reward scenarios.
>
> (2). For the state-level retrieval, Applying retrieval at the state level offers several advantages. First, the state space is compact, which enables efficient retrieval. Second, the fine granularity of states allows flexible recombination while preserving semantic consistency. Thirdly, larger composition units substantially shrink the search space and reduce the likelihood of finding suitable matches. For these reasons, we adopt a state-level retrieval scheme.
>
> (3). To analyze the effect of conditioning on additional target information in trajectory forecasting, we denote the subsequent trajectory generated for planning as a random variable $\tau$, the current state as  $s_t$, and the retrieved target state as  $s_g$. The predictive uncertainty of the trajectory given the current state is quantified by the conditional entropy $H(\tau \mid s_t)$, where a higher entropy reflects a higher degree of uncertainty and thus lower predictive reliability. When the goal information $s_g$  is incorporated as an additional conditioning variable, the uncertainty becomes $H(\tau \mid s_t, s_g)$. Applying the chain rule of conditional entropy to the joint variable pair $(\tau, s_g)$ yields two equivalent decompositions $H(\tau, s_g \mid s_t) = H(\tau \mid s_t) + H(s_g \mid \tau, s_t) = H(s_g \mid s_t) + H(\tau \mid s_t, s_g)$. Rearranging terms leads to $H(\tau \mid s_t) - H(\tau \mid s_t, s_g) = H(s_g \mid s_t) - H(s_g \mid \tau, s_t)$, which matches the definition of conditional mutual information $I(\tau; s_g \mid s_t)$. Since conditional mutual information is always non-negative, i.e.,  $I(\tau; s_g \mid s_t) \ge 0$ , it follows that $H(\tau \mid s_t) \ge H(\tau \mid s_t, s_g)$. This result demonstrates that conditioning on the retrieved target state preserves or decreases the entropy of the trajectory distribution. Therefore, whenever $I(\tau; s_g \mid s_t) > 0$, the introduction of  $s_g$  provides additional relevant information that reduces uncertainty and leads to more accurate and reliable trajectory prediction (i.e.the generation of the subsequent trajectory for planning).

---

> ### Author Response · Authors · 2025-11-22
> **Responses to Reviewer Wzph(Q4-Q5)**
>
> **Q4.** How sensitive is RAD to the accuracy of the value-based ranking in target retrieval? Would errors in return estimation significantly degrade performance?
>
> **A4.** We thank the reviewer for raising this important point.
>
> In our implementation, the value ranking used for retrieval is based on the ground-truth returns (RTG) stored in the offline dataset, rather than a learned value estimator. Therefore, the reported results are not affected by return-estimation errors.
>
> To understand whether imperfect ranking would harm performance, we conducted an additional study on AntMaze by intentionally degrading the ranking quality. Specifically, in the final step of the TS module, instead of always selecting the top-1 state, we formed candidate sets of size 1, 3, 5, and 7 and randomly picked one target from each set. This simulates scenarios where the retrieved target is not always the optimal one. The results are shown below:
>
>
> |Environment|Top-1|Top-3|Top-5|Top-7|
> |-|:-:|:-:|:-:|:-:|
> |Antmaze-medium-play|$86.7$|$85.3$|$85.3$|$72.0$|
> |Antmaze-large-play|$80.0$|$73.3$|$70.0$|$66.7$|
> |Antmaze-medium-diverse|$93.3$|$86.7$|$84.3$|$83.3$|
> |Antmaze-large-diverse|$73.3$|$62.0$|$58.7$|$62.0$|
>
> As the candidate set grows larger, the noise in the retrieval ranking increases, and the performance shows a gradual downward trend. This behavior is expected: when the retrieved target state is not necessarily the optimal one, the guidance provided to the planner becomes weaker, leading to reduced success rates. More importantly, however, this degradation is gradual rather than catastrophic. Comparing these results against the baselines in Table 1 of the paper, we observe that even the worst Top-7 performance remains competitive in most environments. For example, in AntMaze-Medium-Diverse, the Top-7 setting still achieves 83.3, ranking among the top three methods. Therefore, although imperfect ranking introduces some negative effects on the performance, RAD can still benefit from the retrieved target even when it is suboptimal, as long as the retrieved state lies within a reasonably high-value region.
>
> **Q5.** Can RAD handle multi-modal retrieval results (e.g., when several high-return trajectories exist but lead to different goals)?
>
> **A5.** We thank the reviewer for this question. RAD is capable of handling multi-modal retrieval results. In our implementation, the retrieval pool is filtered to include only trajectories consistent with the current task objective. Within this filtered set, the TS module selects the target state relevant to the current task and generates the corresponding sub-trajectory. Consequently, even when multiple high-return trajectories exist, RAD can effectively handle these multi-modal candidates while maintaining consistency with the task objective.

---

> ### Author Response · Authors · 2025-11-25
> **Do Our Responses Resolve Your Main Concerns?**
>
> Dear Reviewer Wzph,
>
> I hope this message finds you well. We have provided detailed responses to your insightful comments, which we hope address them satisfactorily. Some of the key clarifications include the following:
>
> 1. **Generalization evaluation**. We conducted additional OOD tests to evaluate whether learned policies generalize to unseen states. Results show that our method still outperforms existing baselines.
> 2. **Clarification of design motivation and contribution**. Although our framework reuses components from Diffuser and DiffuserLite, the main contribution is a model-agnostic retrieval-guided planning paradigm that improves decision-making in OOD and sparse-reward scenarios.
> 3. **Sensitivity analysis of retrieval ranking**. We evaluated the effect of imperfect ranking. Performance gradually decreases with increased ranking noise but remains competitively strong.
>
> We sincerely hope that these clarifications have resolved your concerns.  If so, we would greatly appreciate it if you could consider updating your evaluation score in light of these clarifications. If there are any additional points or feedback you would like us to consider, please let us know. Your insights are invaluable to us, and we are eager to address any remaining issues to improve our work.
>
> Thank you for your time and effort in reviewing our paper.
>
> Best regards,
>
> The Authors

---

### Author Response · Authors · 2025-12-03
**Final summarization of author-reviewer discussion**

Dear Area Chair,

Thank you for taking the time to review our work and for your service to the offline RL community during this challenging period. We summarize the key context of our rebuttal below. While the discussion phase saw limited engagement from some reviewers, we have responded to all raised points as thoroughly as possible.

### Core Contributions and Practical Impact
The core idea of our work is to introduce a simple yet effective retrieval mechanism on planning-based offline RL, transforming conventional planning (typically target-agnostic) into a high-return-guided and target-oriented process. Specifically, RAD retrieves high-return and reachable states from the offline dataset as targets, and uses a generative model to synthesize sub-trajectories conditioned on these targets for planning.  Since these planning targets lie in high-return regions, guiding the agent toward them enables it to follow the corresponding high-quality action patterns, which in turn significantly improves policy generalization, especially in out-of-distribution (OOD) and sparse-reward settings. It is worth emphasizing that our proposed retrieval-guided paradigm is model-agnostic and can be seamlessly integrated with any generative-model-based planner.

### Strengths

The reviewers evaluated our work positively and identified several notable strengths.

- Clearly motivates the integration of state retrieval into offline RL as adaptive guidance for policy improvement. `Wzph`, `wzXn`


- Novel design that adaptively retrieves targets (not static augmentation), which aids planning in sparse-reward and out-of-distribution tasks. `6yLX`, `woTf`

- The “target-then-plan” decomposition (Target Selection → Step Estimation → Planning) is conceptually clear, and "may be applied to many sequence-modeling or model-free offline baselines". `6yLX`, `woTf`

- The experiment done in the paper is quite comprehensive, including a solid suite of baselines, visualization, ablations, and distribution-shift study. “The actual results on the tasks perform quite well. In general this method is doing much better than the baselines.” `Wzph`, `6yLX`, `woTf`

### Discussion points

The main concerns of reviewers and the corresponding responses are summarized as follows:
- Reviewers Wzph and 6yLX concern the theoretical guarantees of the retrieval mechanism.
  - We explicitly explained why retrieving target states can improve the reliability of generative-model-based planning, and provided a theoretical analysis in Appendix F.

- Reviewers 6yLX and wzXn raise the concerns about the differences from prior work.
  - We have provided a detailed response and added a clarification in the Appendix G regarding the distinction between RAD and data augmentation methods. RAD is an offline RL algorithm that dynamically retrieves high-return target states for planning and decision making, rather than expanding the training dataset as typical data augmentation methods do. Theoretically, the data generated by data augmented methods could further be used to train RAD, making the two approaches compatible and organically combinable for more efficient decision-making.

- Reviewers Wzph and  6yLX are concerned about impact of ranking noise on policy stability.
  - We have conducted  extensive experiments to evaluate whether ranking noise would harm performance, and provided the results in both response and  Appendix E.5. The results show that although ranking noise introduces some negative effects, RAD can still benefit from the retrieved targets as long as they lie within a reasonably high-value region, even if the selected target is not the optimal one.

- Reviewers woTf and wzXn ask for further discussion of inference cost.
  - We have measured the total time taken by RAD to generate a single action, and provide detailed results in response and Appendix E.1. Given the performance improvement of RAD over baseline, we believe that this additional latency is justified.

 We sincerely hope these clarifications help you fully assess the technical value, novelty, and potential impact of our work. We greatly appreciate your time and expert guidance during this review. Given the broad set of additional analyses we have conducted in this rebuttal, we kindly ask the AC to review our revised paper.

---

### Meta-Review · Area_Chair_ZYdk · 2026-01-03

**Summary:**

The reviewers’ concerns center on limited conceptual novelty, insufficient theoretical grounding, and methodological weaknesses. While the paper integrates retrieval with diffusion-based planning for offline RL, multiple reviewers noted that the core ideas substantially overlap with prior trajectory-stitching and model-based offline RL methods, and that the contribution is primarily an incremental integration rather than a clear conceptual advance. Theoretical justification remains weak and largely heuristic, offering limited insight into reachability, stability, or policy improvement guarantees. Reviewers also raised concerns about experimental rigor (e.g., sensitivity to backbone choice, initially missing statistical reporting) and clarity of method description, several of which required post-hoc clarification. Although the authors provided detailed responses and additional experiments, these did not fundamentally change reviewers’ assessments. These concerns led to the recommendation to reject.

**Reviewer Concerns:**

**Concerns Partially or Largely Addressed**：

1. **Clarity and presentation issues**:  The authors corrected multiple typos, notation inconsistencies, and ambiguities in the method description, and clarified policy execution (closed-loop operation, action extraction), improving overall readability.

2. **Experimental reporting and statistics** (6yLX, woTf):  The rebuttal added variance/confidence information and average scores, addressing concerns about insufficient statistical reporting and unusual table highlighting criteria.

3. **Sensitivity to imperfect target ranking** (Wzph):  Additional ablation studies were included to analyze ranking noise in target retrieval, showing gradual rather than catastrophic performance degradation.

**Concerns That Remains**：

1. **Limited conceptual novelty** (6yLX, wzXn):  Reviewers remain unconvinced that RAD constitutes a fundamentally new algorithmic contribution beyond existing trajectory-stitching, diffusion-based planning, and offline RL methods. The rebuttal clarified positioning but did not materially change this assessment.

2.  **Weak theoretical grounding** (6yLX):  The information-theoretic argument is generic and does not address core concerns about reachability guarantees, policy stability, or long-horizon planning reliability under imperfect retrieval.

3. **Overall significance relative to venue bar**:  Despite improved clarity and additional experiments, reviewer (wzXn) with below-threshold scores explicitly maintained their evaluations.

**Reviewer Scores:**

- **Reviewer Wzph (score: 6)**:  No change.  could potentially drop after reading other reviewers' comments.

- **Reviewer 6yLX (score: 4)**:  No change.

- **Reviewer woTf (score: 2)**:  No change.

- **Reviewer wzXn (score: 4)**:  No change.

---

### Decision · Program_Chairs · 2026-01-26

Reject